

# Dynamical instantons and activated processes
# in mean-field glass models

**Valentina Ros[1,2⋆], Giulio Biroli[2] and Chiara Cammarota[3,4]**

**1** Université Paris-Saclay, CNRS, LPTMS, 91405, Orsay, France
**2** Laboratoire de Physique de l'Ecole normale supérieure, ENS, Université PSL, CNRS,
Sorbonne Université, Université Paris-Diderot, Sorbonne Paris Cité, Paris, France
**3** Dip. Fisica, Universitá "Sapienza", Piazzale A. Moro 5, I-00185, Rome, Italy
**4** Department of Mathematics, King's College London, Strand London WC2R 2LS, UK

⋆ valentina.ros@universite-paris-saclay.fr

## Abstract

We focus on the energy landscape of a simple mean-field model of glasses and ana-
lyze activated barrier-crossing by combining the Kac-Rice method for high-dimensional
Gaussian landscapes with dynamical field theory. In particular, we consider Langevin dy-
namics at low temperature in the energy landscape of the pure spherical $p$-spin model.
We select as initial condition for the dynamics one of the many unstable index-1 saddles
in the vicinity of a reference local minimum. We show that the associated dynamical
mean-field equations admit two solutions: one corresponds to falling back to the origi-
nal reference minimum, and the other to reaching a new minimum past the barrier. By
varying the saddle we scan and characterize the properties of such minima reachable
by activated barrier-crossing. Finally, using time-reversal transformations, we construct
the two-point function dynamical instanton of the corresponding activated process.

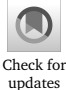

# 1 Introduction

Rough high-dimensional energy landscapes are central in many different contexts. In physics, they are one of the key ingredients of the theory of glasses, and more generally of disordered systems [1]. In computer science, they are studied to characterize algorithmic phase transitions for inference and signal processing [2], and they have attracted a lot of attention in the field of deep neural networks [3]. In biology, they appear in analysis of evolution and in the study of protein folding [4, 5].

In all these disparate contexts the system under study explores a rough landscape by stochastic dynamics, and the main aim is to characterize the complex dynamical behavior that ensues

from it. The mean-field theory of glasses and spin-glasses has been instrumental in this respect. It provided the first quantitative analysis of rough high-dimensional landscapes [6], in particular of the number and the properties of the critical points, and of the associated stochastic dynamics [7]. It was shown theoretically that starting from a random high-energy initial condition (corresponding to a high enough temperature in an equilibrium setting), mean-field glass models very slowly approach metastable states, which are typically the most numerous ones and are marginally stable, meaning that their Hessian matrix is characterized by arbitrary small eigenvalues [8]. The intuitive explanation of this phenomenon is that these metastable states, called the threshold states [1], are the most numerous and the wider ones, and hence naturally correspond to the largest basins of attraction. This paradigmatic behavior has been applied and transposed with success in a variety of contexts in the last twenty years, in particular to explain the glassy dynamics of three-dimensional interacting particle systems (super-cooled liquids, colloidal glasses, etc.) [1].

Calling $N$ the dimension of the energy landscape, which in physics contexts is proportional to the number of degrees of freedom, mean-field glass models display two dynamical regimes: a *slow descent regime* that corresponds to time-scales that do not diverge with $N$ in which the system approaches (or more precisely ages toward) the threshold (or more generally, the marginally stable) states. An *activated regime* in which the system jumps over increasingly larger barriers and is able to explore fully the energy landscape. To observe such activated processes in mean-field models one has to probe the dynamics on time-scales which are exponentially large in $N$ since the barriers between low energy metastable states scale as $N$ [10–13]. Whereas a theory of the first dynamical regime has been progressively developed in the last twenty years, constructing a theoretical framework to understand the second one remains an open problem—a central one in many of the contexts in which rough energy landscapes play a role.

The main reason for this state of affairs is that activated dynamics is well understood mainly in low dimensional cases, where the number of minima and of saddles connecting them is *finite* and possibly small. The standard methods to tackle this problem were developed quite independently in statistical physics to analyze the phenomenon of nucleation at first-order phase transitions [14], in quantum field-theory for tunneling between degenerate vacua (where Planck's constant plays the role of temperature) [15], and in probability theory [16]. On the contrary, rough high-dimensional energy landscapes are characterized by *diverging* (exponentially in $N$) number of paths that connect a *diverging* number of metastable states. In this case standard frameworks are not adapted, and new ideas and methods are needed.

The main technical difficulty in establishing a theory of activated processes for mean-field glassy systems and high-dimensional rough landscapes is that the correct order parameter that describes glassy dynamics is the correlation function between two different times [7]. This introduces an additional degree of difficulty with respect to the standard setting in phase transitions, where the order parameters are typically one-time (or point) functions. In consequence, contrary to known situations in which to describe an activated process one has to find the rare trajectory, called instanton, that connects two minima and that corresponds to the optimal change of the one-point function corresponding to the order parameter [17], in this case one has to find the instanton on a two-point function. This is a quite different, less intuitive and more complex mathematical object. Henceforth, in order to highlight this difference, we will call it *dynamical instanton*. Although some results have been given in the past litera-

---

[1]Here and henceforth, with "threshold states" we refer to those marginally stable states that are reached asymptotically by Langevin dynamics initialized at high enough (*e.g.*, infinite) temperature. In models in which the marginally stable states are distributed over an extensive range of energies, the "threshold" ones have the energy density of the most numerous marginal states. Marginal states of different energy can also be reached asymptotically by the dynamics, as shown for instance in Ref. [9], provided that the system is initialized at lower temperatures.

ture [18–21], the problem of finding the dynamical instanton corresponding to the activated jump out of a given minimum of the energy landscape of a mean-field glass model remains a largely unsolved challenge. Here we provide the first computation of such dynamical instanton and characterize the properties of the new minima reached after barrier-crossing.

In order to achieve this goal we make use of the results we obtained recently on the number of the stationary points constrained to be at fixed overlap $q$ (or distance $d$) from a given minimum in a prototypical energy landscape [22,23] (see also [24] for a related analysis). These studies showed that, given an arbitrary local minimum $s_1$ of the energy landscape with energy density $\epsilon_1$, the landscape in its vicinity is populated by index-1 saddles, that constitute available escape states when the system is trapped in $s_1$. By extending to dynamics the theoretical framework developed for the high-dimensional Kac-Rice method (see also [25]), we derive dynamical equations describing the evolution of the system conditioned to start from such unstable saddles as initial states. By analyzing these equations analytically and integrating them numerically, we show that they admit two solutions which are associated to the descents toward the two minima reachable from the saddle. In this way, we map out the first geometrical properties of the Morse complex, i.e. we characterize all the local minima that are connected to the original reference one through index-1 saddles (as illustrated in Fig. 1). We then resort to dynamical field theory and to the time-reversal property of stochastic dynamics to construct the dynamical instantons for the two-point functions. The two dynamical solutions discussed above are used as building blocks: the part of the dynamical instanton associated to the ascent of the system from the original minimum to the nearby saddle is obtained through the time reversal of the relaxation path from the saddle down to the minimum [19,20]. We then combine this contribution with the one corresponding to the descent from the saddle to the new minimum to finally obtain the shape of the dynamical instanton, see Fig. 4.

In the following section a summary of the state-of-the-art and our main contributions is presented, we will then expose in details our analysis.

## 2 Summary of results

### 2.1 Model and state-of-the-art

We focus on the energy landscape associated to the $p$-spin spherical model:

$$\mathcal{E}[s] = -\sum_{i_1,\cdots,i_p} J_{i_1\cdots i_p} s_{i_1} \cdots s_{i_p}, \tag{1}$$

defined at each point $s = (s_1,\cdots,s_N)$ of an $N$-dimensional sphere, $s\cdot s = N$. The couplings $J_{i_1\cdots i_p}$ are independent Gaussian random variables with zero average and variance $\langle J_{i_1\cdots i_p}^2 \rangle = 1/2p!N^{p-1}$, and are symmetric under permutations of the indexes. The functional (1) has been the subject of an extensive amount of research devoted to understanding its statistical properties, which started with the earlier investigations [26–31] and culminated in the most recent results [22, 23, 32, 33]. These works highlighted a peculiar organization of the landscape stationary points in terms of their energy density $\epsilon = \mathcal{E}/N$ and of their stability: while at large value of the energy the landscape is dominated by saddles with a huge index (i.e., number of unstable directions), the local minima and low-index saddles concentrate at the bottom of the landscape, below a critical *threshold* value of the energy density $\epsilon_{\text{th}}$. Their number $\mathcal{N}_k(\epsilon)$ ($k$ being the index) is exponentially large in $N$, its typical value being governed by a positive complexity $\Sigma_k(\epsilon) = \lim_{N\to\infty} \langle \log \mathcal{N}_k(\epsilon) \rangle / N$ that decreases with $k$. The high-dimensionality of configuration space entails that most of these low-energy minima and saddles are orthogonal to each others on the sphere, i.e. they normalized overlap

$q(s, s') = \lim_{N \to \infty} s \cdot s'/N$ is typically equal to zero.

In order to find the escape paths from a given minimum, one needs to perform a more thorough analysis. In particular, it is important to scan the landscape in the vicinity of any of its stationary points. This information is accessible via large deviation techniques by computing the complexity of the stationary points constrained to be at fixed, non-zero overlap from the reference stationary point [22–24].

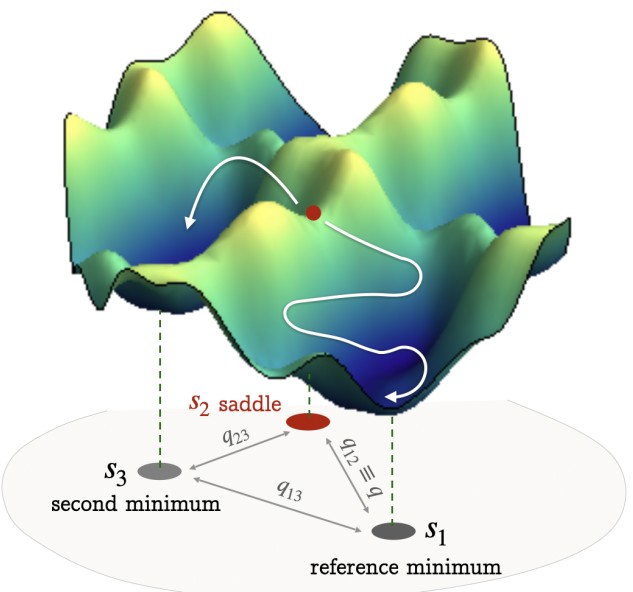

Figure 1: Pictorial representation of the landscape with a pair of local minima connected by a saddle $s_2$. The white lines represent the dynamical evolution of the system conditioned to start from the saddle as an initial condition.

In Ref. [22] we computed the complexity of the typical stationary points that are found in the vicinity of a reference minimum, extracted uniformly from the ensemble of minima having a given energy density larger than the ground state $\epsilon_{gs}$ and smaller than the threshold $\epsilon_{th}$. Henceforth we denote with $s_1$ the reference minimum, and with $s_2$ a stationary point at overlap

$$s_1 \cdot s_2 = N q_{12} \equiv N q, \tag{2}$$

from it. We let $\epsilon_1, \epsilon_2$ be the corresponding energy densities, and $\Sigma(\epsilon_2, q|\epsilon_1)$ the complexity of the stationary points at energy $\epsilon_2$. The results for a representative value of energy $\epsilon_1$ of the reference minimum are summarized in Fig. 2. The stationary points that are closer to the reference minimum (i.e., at larger overlap) typically appear at an overlap that we denote with $q_M$, and are at high energy density (equal to $\epsilon_{th}$ in the case of Fig. 2). At each overlap smaller than $q_M$ we find an exponentially large number of stationary points ($\Sigma > 0$), with energy densities $\epsilon_2 > \overline{\epsilon}(q|\epsilon_1)$. The lower bound $\overline{\epsilon}(q|\epsilon_1)$ corresponds to the energy of the deepest stationary points at overlap $q$: their complexity is exactly zero. For any fixed $\epsilon_2$ smaller than $\epsilon_{th}$ (see for instance the dashed arrow in Fig. 2), the closest stationary points with that energy are found at an overlap $q_m(\epsilon_2)$, and are index-1 saddles: their Hessian has an eigenvalue density with a positively supported bulk (like for minima), plus an isolated eigenvalue that is separated from the bulk, and negative. The eigenvector associated to that eigenvalue has a macroscopic projection along the direction connecting $s_2$ and $s_1$ in configuration space, indicating that the saddle $s_2$ is unstable in a direction that 'points' towards the minimum $s_1$. This remains true decreasing the overlap, up to a value $q_{ms}(\epsilon_2)$ where a transition to minima occurs. This is

the overlap at which the curve $\epsilon_{\mathrm{ms}}(q|\epsilon_1)$ intersects the given $\epsilon_2$: the points at this overlap are marginally stable index-1 saddles, with one flat mode (the isolated eigenvalue is exactly equal to zero). For smaller overlaps the stationary points are minima; the closest ones are still correlated to the minimum $s_1$ (dashed gray region in Fig. 2), since their Hessian still exhibits an isolated eigenvalue, that is nevertheless positive. Eventually, for even smaller $q$ the points become minima that are totally uncorrelated to $s_1$. At $q = 0$, we recover the unconstrained complexity of local minima. Therefore, all the stationary points enclosed in the violet region of Fig. 2 are typically saddles that are geometrically connected to the reference minimum in configuration space. Their complexity is shown in the same figure. Among them, the deepest ones have parameters $q^*(\epsilon_1), \epsilon_2^*(\epsilon_1)$ that correspond to the intersection between the curves $\epsilon_{\mathrm{ms}}(q|\epsilon_1)$ and $\overline{\epsilon}(q|\epsilon_1)$.

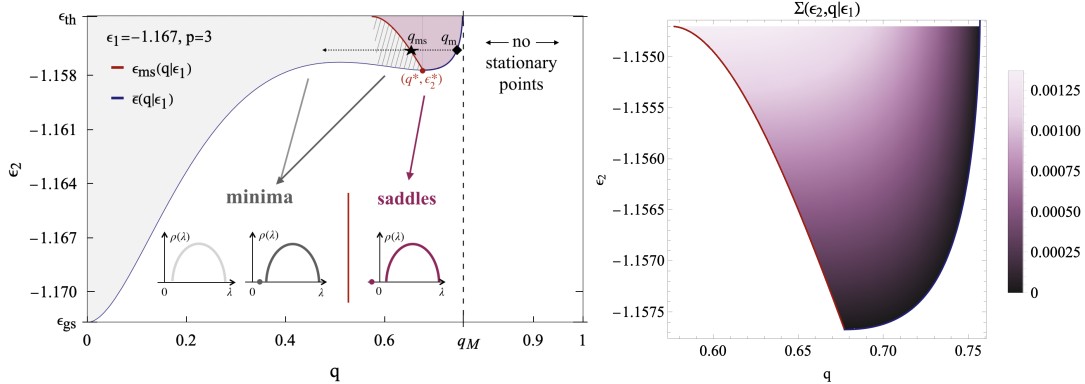

Figure 2: *Left.* The colored regions identify the range of energy densities $\epsilon_2$ of the stationary points found at overlap $q$ with a minimum of energy density $\epsilon_1 = -1.167$: the violet area corresponds to index-1 saddles, the gray one to minima that are either correlated (dashed area) or uncorrelated to the reference one. The eigenvalue density $\rho(\lambda)$ of the Hessian matrices at the stationary points is sketched at the bottom of the plot. *Right.* Color plot of the complexity of the index-1 saddles as a function of their energy and overlap $q$ with the reference minimum of energy $\epsilon_1 = -1.167$.

## 2.2 The landscape in the vicinity of a local minimum

All the saddles lying in the vicinity of the reference minimum $s_1$, and corresponding to the violet region in Fig. 2, represent possible escape states for the system trapped in the local minimum. In this work we study *where* the gradient descent dynamics starting from these saddles lands in the energy landscape. By developing a theoretical framework that combines the Kac-Rice method and dynamical mean-field theory, we obtain the dynamical equations that allow to characterize the minima $s_3$ that are connected to the reference one $s_1$ through one of the saddles $s_2$ lying in its vicinity, see Fig. 1. These minima are the states that the system can reach if it manages to escape from $s_1$ through one of the surrounding saddles. The connectivity of $s_1$ in configuration space can thus be characterized by studying the energy density $\epsilon_3$ and the overlap $Nq_{13} = s_1 \cdot s_3$ of the minima $s_3$ reached asymptotically, as a function of the saddle parameters $q, \epsilon_2$.

Fig. 3 shows the resulting distributions, for a representative value of $\epsilon_1$ as above. Interesting correlations emerge between minima and saddles: at fixed energy $\epsilon_2$ of the saddle, those at higher overlap (i.e., those closer to the reference minimum) are more optimal, as they allow to reach minima that lie deeper in the landscape, and at furthest distance from the reference minimum. Upon changing the energy of the saddle, one discovers that there exists a trade-off between energy and overlap: the saddles that connect the reference minimum to the deeper

ones are not the same ones that connect it to the further ones, thus allowing to explore a larger portion of configuration space. In particular, the saddles at $q^*, \epsilon_2^*$ that correspond to the minimal energy barrier are optimal in terms of energy of $s_3$, but not in terms of its overlap $q_{13}$. Overall, we see that both the range in energy and in overlap of the connected minima is rather limited: escaping through these saddles, the system reaches minima that are highly correlated from the reference one. We comment on the implications of this on the dynamics in Sec. 7, and refer to Sec. 4 for a more detailed analysis of the asymptotic solutions of the dynamical equations.

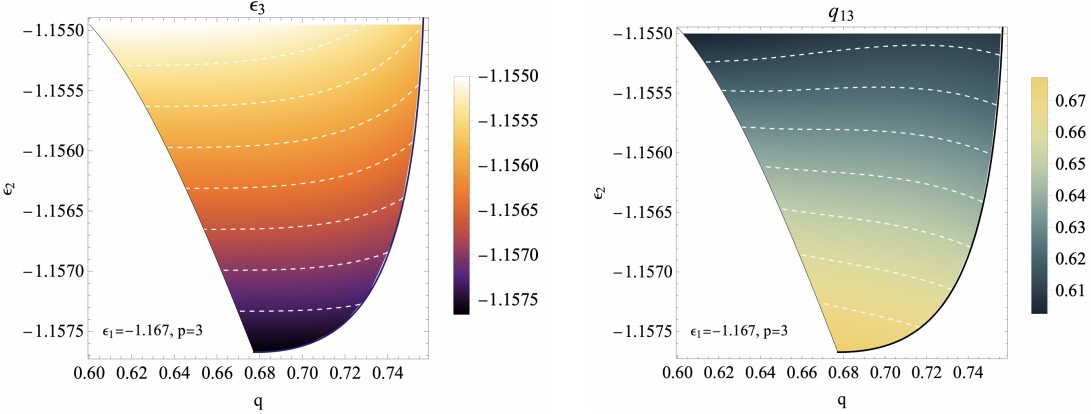

Figure 3: *Left.* Energy density $\epsilon_3$ of the minimum reached asymptotically by the dynamics starting from an index-1 saddle at energy $\epsilon_2$ and overlap $q$ with the reference minimum having energy $\epsilon_1 = -1.167$. *Right.* Overlap $q_{13}$ between the reference minimum and the one reached asymptotically by the dynamics starting from the saddle at energy $\epsilon_2$ and overlap $q$ with the reference minimum. The white dashed lines in both figures are level curves.

## 2.3 Activated processes and dynamical instantons

As already stressed in the introduction, one of the main aims of this work is to obtain the dynamical instanton that corresponds to the activated process associated to the escape from a given minimum $s_1$ towards a new minimum $s_3$, see Fig. 1. In the theory of stochastic processes, instantons are in general obtained as special extremal solutions of a large deviation functional[2] [16, 34]. In the case of mean-field spin glasses the corresponding mathematical object is a functional of the two-point functions [35, 36]. Although, in principle one could look for dynamical solutions by extremizing this functional and imposing suitable boundary conditions in time, in practice analyzing the corresponding equations represents a formidable challenge. No numerical solution has been obtained yet. On the analytic side, despite the results in [18–21] the problem remains largely open, mainly due to the lack of intuition on the kind of solution one is looking for. The only case in which a dynamical instanton has been fully worked out is in the study of finite-time metastable states where periodic boundary condition in time are enforced [36], which is however quite a different situation with respect to the one we are interested in here.

In the following we show how to obtain the dynamical instanton corresponding to the activated process sketched in Fig. 1. The resulting shape of the two-point correlation function is shown in Fig. 4. It displays three time regimes: the first one corresponding to the ascent from the minimum $s_1$ to the saddle $s_2$, the second one corresponding to the approach and the

---

[2]In general, this large deviation functional quantifies the probability of the stochastic process to reach in a given time a certain class of configurations.

departure from the saddle, and the final one associated to the descent towards the new minimum $s_3$. Since the basic objects is a symmetric two-time functions, $c(t, t')$, this leads to six different time-sectors and six different behaviors for the correlation function (depending on which of the three regimes the times $t$ and $t'$ belong to). As sketched in Fig. 1, this solution is obtained conditioning the system to escape from $s_1$ through one particular, chosen index-1 saddle $s_2$: it therefore does not represent the most general escape process, that should be obtained averaging over all possible dynamical trajectories connecting the two local minima, possibly through different saddles. We therefore expect that the $c(t, t')$ in Fig. 4 represents a special solution of more general dynamical equations, obtained extremizing a large deviation functional as mentioned above. Despite being a special case, the explicit form of the dynamical instanton associated to the simple activated process in Fig. 1 is instrumental in finding instantons associated to more complex relaxation processes, in particular to equilibrium relaxation. We shall get back to this issue in the conclusion.

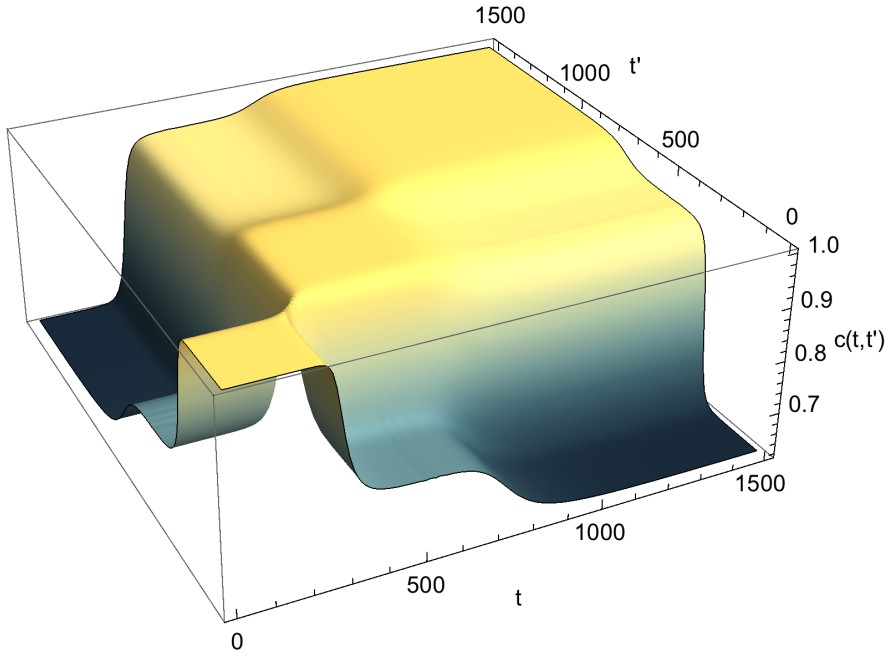

Figure 4: Representation of the correlation function $c(t, t')$ along the reconstructed (see Sec. 6) instantonic solution that links a reference minimum ($s_1$) at energy $\epsilon_1 = -1.167$ to a neighboring minimum ($s_3$) reached through a saddle ($s_2$) at energy $\epsilon_2 = -1.1555$ and with overlap $q = 0.75$ with $s_1$. The plot shows correlation equal to one on the diagonal and plateaux on other three different levels, corresponding to the overlaps $q = 0.75$ between $s_1$ and $s_2$, $q_{23} = 0.957$ between $s_2$ and $s_3$, and $q_{13} = 0.619$ between $s_1$ and $s_3$.

## 3 Self-consistent dynamical equations describing the escape from a saddle

In this section we derive the equations describing the system evolution with specific initial conditions, that correspond to being in a saddle at a fixed distance of a given local minimum of the energy landscape.

We remark that dynamical equations with constrained initial conditions for the $p$-spin spher-

ical model have been derived in simpler settings, see for instance Refs. [25, 37, 38]. In particular, Ref. [37] studies the exponential relaxation of the system initialized within one of the metastable states that contribute to the Boltzmann measure in the so called dynamical phase, at temperatures between the static and the dynamic transition temperatures. In Ref. [38] and in the more recent [25] the overlap between the initial condition of the dynamics and a thermalized condition in the same temperature range is also enforced to take a fixed, non-zero tunable value.

The approach we present below goes one step further since we condition on the initial condition $s_2$ to be itself a stationary point, beside conditioning on its energy density and on the overlap with the reference minimum $s_1$. From the technical point of view our approach combines the Kac-Rice method developed to study critical points of high-dimensional rough landscapes [32, 39] with dynamical field theory [34, 35].

### 3.1 The dynamical action with constrained initial conditions

Let $s(t)$ denote the spin configuration at time $t$, and let $\mathcal{E}[s^t]$ be the time-dependent energy field evaluated at $s(t)$:

$$\mathcal{E}[s^t] = -\sum_{i_1,\cdots,i_p} J_{i_1\cdots i_p} s_{i_1}(t)\cdots s_{i_p}(t). \tag{3}$$

The vector $s(t)$ is obtained as a solution of the Langevin equation:

$$\frac{ds_i(t)}{dt} = -\frac{\delta \mathcal{E}[s^t]}{\delta s_i(t)} - z(t)s_i(t) + \xi_i(t), \tag{4}$$

where $\xi_i(t)$ is white noise with correlations

$$\langle \xi_i(t)\xi_j(t')\rangle = \alpha \delta_{ij}\,\delta(t-t'), \quad \alpha \geq 0, \tag{5}$$

$z(t)$ is a Lagrange multiplier that enforces the spherical constraint $s(t)\cdot s(t) = N$, and

$$\frac{\delta \mathcal{E}[s^t]}{\delta s_i(t)} = -p\sum_{i_2,\cdots,i_p} J_{ii_2\cdots i_p} s_{i_2}(t)\cdots s_{i_p}(t). \tag{6}$$

The strength of the noise $\alpha$ is equal to twice the temperature $T$. With this normalization, typically $\mathcal{E}[s^t] \sim \sqrt{N}$. We assume that the dynamics has a specified initial condition $s(0) = s_2$, with corresponding energy:

$$E_2 = N\epsilon_2 \equiv \mathcal{E}[s_2] = -\sum_{i_1,\cdots,i_p} J_{i_1\cdots i_p} (s_2)_{i_1}\cdots(s_2)_{i_p}. \tag{7}$$

The dynamical generating functional corresponding to the stochastic evolution (4) and obtained integrating over the noise reads:

$$Z_D(s_2) = \int_{s(0)=s_2} \mathcal{D}s^t \mathcal{D}\hat{s}^t \exp\left\{\sum_{i=1}^{N}\int_0^\infty dt\,\hat{s}_i(t)\left[\frac{\alpha}{2}\hat{s}_i(t) - \frac{ds_i(t)}{dt} - z(t)s_i(t) - \frac{\delta\mathcal{E}[s^t]}{\delta s_i(t)}\right]\right\}, \tag{8}$$

where $\hat{s}_i(t)$ are auxiliary fields[3] and we highlighted the dependence on the initial condition of the dynamical evolution.

---

[3]This is obtained exponentiating the delta function imposing the dynamical constraint (4), and performing the rotation $i\hat{s}_j(t) \to -\hat{s}_j(t)$. The Itô prescription is used, implying that the Jacobian is equal to one.

### 3.1.1 Implementing the initial conditions

We aim at averaging the above dynamical functional over all possible initial conditions $s_2$ that are stationary points found in the vicinity of some local minimum $s_1$ of the energy landscape, having energy:

$$E_1 = N\epsilon_1 \equiv \mathcal{E}[s_1] = -\sum_{i_1,\cdots,i_p} J_{i_1\cdots i_p}(s_1)_{i_1}\cdots(s_1)_{i_p}. \tag{9}$$

We assume that the two stationary points are at overlap $Nq = s_1 \cdot s_2$, and define the following average over the initial conditions:

$$\mathbb{E}_q[\otimes] = \int ds_1 \int ds_2\, \delta(s_1 \cdot s_2 - Nq)\, \frac{f_{\epsilon_1}(s_1)}{\mathcal{N}(\epsilon_1)}\, \frac{f_{\epsilon_2}(s_2)}{\mathcal{N}_{s_1}(\epsilon_2, q|\epsilon_1)} \otimes, \tag{10}$$

where the integrals are over the sphere of radius $\sqrt{N}$. The explicit form of the measure is given by:

$$f_\epsilon(s) = \prod_{\alpha=1}^{N-1} \delta\left(\sum_{i=1}^N e_i^\alpha[s] \frac{\partial\mathcal{E}[s]}{\partial s_i}\right) \delta(\mathcal{E}[s] - N\epsilon)\, |\det\mathcal{H}[s]|. \tag{11}$$

This measure encodes the spherical constraint, since $e^\alpha[s]$ with $\alpha = 1,\cdots,N-1$ are unit vectors spanning the tangent plane of the sphere at the point $s$, and $\mathcal{H}[s]$ is the Hessian matrix of the energy functional at the point $s$, which is also projected onto the tangent plane and has components:

$$\mathcal{H}_{\alpha\beta} = e^\alpha[s] \cdot \frac{\partial^2\mathcal{E}[s]}{\partial s^2} \cdot e^\beta[s] - \frac{1}{N}\left(\frac{\partial\mathcal{E}[s]}{\partial s} \cdot s\right)\delta_{\alpha\beta}. \tag{12}$$

The normalization $\mathcal{N}_{s_1}(\epsilon_2, q|\epsilon_1)$ denotes the total number of stationary points of energy density $\epsilon_2$ that are found at overlap $q$ with a fixed stationary point $s_1$ having energy density $\epsilon_1$, whereas $\mathcal{N}(\epsilon_1)$ is the total number of stationary points of energy density $\epsilon_1$. We thus define the generating functional averaged over the initial conditions as:

$$\mathcal{Z}_D = \mathbb{E}_q[Z_D(s_2)]. \tag{13}$$

The interpretation of (10) is as follows: the measure (11) weights uniformly all stationary points at a given value of energy density $\epsilon_1$; for energies below the threshold energy, these stationary points are typically local minima, since the number of saddles of finite index is exponentially suppressed in the dimension $N$ (with respect to the number of minima). Therefore, in the large-$N$ limit the point $s_1$ extracted with such measure will be a local minimum with a high probability, that converges to one as $N \to \infty$. Similarly, the analysis of Ref. [22] reveals that for each choice of $q, \epsilon_2$, the energy landscape is dominated by one specific type of stationary points that are either local minima or index-1 saddles, see Fig. 2. More precisely, any local minimum $s_1$ is typically surrounded by an exponentially large (in the dimension $N$) population of stationary points distributed over a finite range of overlaps $q$; among them, the ones that are at larger overlap $q$ (and thus closer to the minimum) are typically index-1 saddles. This implies that if a stationary point is selected at random among those at large-enough overlap, with probability that converges to one in the large-$N$ limit this point will be an index-1 saddle. By suitably changing the parameters $q, \epsilon_2$ at fixed energy $\epsilon_1$, we can select the initial condition $s_2$ to be either an index-1 saddle or a local minimum. Of course we are particularly interested in the regime of parameters in which the initial condition is an unstable stationary point, i.e. a saddle.

### 3.1.2  Averaging over the random couplings

The generating functional (13) can be averaged over the quenched random couplings $J_{i_1,\cdots,i_p}$. We denote the corresponding average with

$$\langle \mathcal{Z}_D \rangle = \left\langle \int \int ds_1 ds_2 \, \delta(s_1 \cdot s_2 - Nq) \frac{f_{\epsilon_1}(s_1)}{\mathcal{N}(\epsilon_1)} \frac{f_{\epsilon_2}(s_2)}{\mathcal{N}_{s_1}(\epsilon_2, q|\epsilon_1)} Z_D(s_2) \right\rangle. \tag{14}$$

Computing this average is *a priori* non-trivial because of the normalizations in the denominator, which are explicit functions of the random couplings. The computation can be performed exploiting the identity $x^{-1} = \lim_{n\to 0} x^{n-1}$. This naturally leads to a replica calculation, in which higher moments of the quantities $\mathcal{N}(\epsilon_1)$ and $\mathcal{N}_{s_1}(\epsilon_2, q|\epsilon_1)$ have to be determined. As it follows from the results of [22], however, such a replica calculation reproduces the results obtained within the so called annealed approximation, which in this case corresponds to averaging separately the numerators and the denominator of (14):

$$\langle \mathcal{Z}_D \rangle = \frac{\left\langle \int \int ds_1 ds_2 \, \delta(s_1 \cdot s_2 - Nq) f_{\epsilon_1}(s_1) f_{\epsilon_2}(s_2) Z_D(s_2) \right\rangle}{\langle \mathcal{N}(\epsilon_1) \mathcal{N}_{s_1}(\epsilon_2, q|\epsilon_1) \rangle}. \tag{15}$$

We can therefore focus on the average of the numerator, which we denote with:

$$\mathcal{I}(\epsilon_2, q|\epsilon_1) = \int ds_1 ds_2 \, \delta(s_1 \cdot s_2 - Nq) \int_{s(0)=s_2} \mathcal{D}s^t \mathcal{D}\hat{s}^t \, e^{\mathcal{V}_0[s^t, \hat{s}^t]} \mathcal{J}(\epsilon_2, q|\epsilon_1), \tag{16}$$

where

$$\mathcal{J}(\epsilon_2, q|\epsilon_1) = \left\langle f_{\epsilon_1}(s_1) f_{\epsilon_2}(s_2) e^{-\sum_{i=1}^N \int_0^\infty dt \, \hat{s}_i(t) \frac{\delta \mathcal{E}[s^t]}{\delta s_i(t)}} \right\rangle, \tag{17}$$

and

$$\mathcal{V}_0[s^t, \hat{s}^t] = \sum_{i=1}^N \int_0^\infty dt \, \hat{s}_i(t) \left[ \frac{\alpha}{2} \hat{s}_i(t) - \frac{ds_i(t)}{dt} - z(t) s_i(t) \right]. \tag{18}$$

To perform the average over the random couplings, we make use of the following trick: because in the pure $p$-spin model the energy functional is homogeneous, it can be expressed (together with its gradient) in terms of the time-dependent symmetric matrix field $M_{ij}(t)$ defined as:

$$M_{ij}[s^t] \equiv \frac{\delta^2 \mathcal{E}[s^t]}{\delta s_i(t) \delta s_j(t)} = -p(p-1) \sum_{i_3,\cdots,i_p} J_{iji_3\cdots i_p} s_{i_3}(t) \cdots s_{i_p}(t). \tag{19}$$

Indeed, we can write:

$$\frac{\delta \mathcal{E}[s^t]}{\delta s_i(t)} = \frac{1}{p-1} \sum_{j=1}^N M_{ij}[s^t] s_j(t), \qquad \mathcal{E}[s^t] = \frac{1}{p(p-1)} \sum_{i,j=1}^N M_{ij}[s^t] s_i(t) s_j(t). \tag{20}$$

The matrix field (19) is symmetric and random, with a Gaussian statistics induced by the couplings. The covariance of the field evaluated along two fixed different dynamical trajectories $s_a(t), s_b(t')$ is given by:

$$\langle M_{ij}[s_a^t] M_{kl}[s_b^{t'}] \rangle = \frac{p(p-1)}{2N} (\delta_{ik}\delta_{jl} + \delta_{il}\delta_{jk}) \left( \frac{s_a(t) \cdot s_b(t')}{N} \right)^{p-2} + \frac{p(p-1)(p-2)}{2N^2} \times$$

$$\times \left\{ [s_b(t')]_i \left( \delta_{jl}[s_a(t)]_k + \delta_{jk}[s_a(t)]_l \right) + [s_b(t')]_j \left( \delta_{il}[s_a(t)]_k + \delta_{ik}[s_a(t)]_l \right) \right\} \left( \frac{s_a(t) \cdot s_b(t')}{N} \right)^{p-3}$$

$$+ \frac{p(p-1)(p-2)(p-3)}{2N^3} [s_b(t')]_i [s_b(t')]_j [s_a(t)]_k [s_a(t)]_l \left( \frac{s_a(t) \cdot s_b(t')}{N} \right)^{p-4}. \tag{21}$$

As a consequence, the average over the random couplings can be equivalently re-written as an average over this matrix field. This is convenient as it allows us to account for the constraints in the initial condition of the dynamics (encoded in the measure (11)) in a straightforward way, given that for both $s_a$ with $a = 1, 2$ we can write the energy and the gradient in terms of the matrix field:

$$\mathcal{E}[s_a] = \frac{s_a \cdot M[s_a^{t=0}] \cdot s_a}{p(p-1)} = N\epsilon_a, \qquad \sum_{i=1}^{N} e_i^\alpha[s_a] \frac{\partial \mathcal{E}[s_a]}{\partial (s_a)_i} = \frac{e_i^\alpha[s_a] \cdot M[s_a^{t=0}] \cdot s_a}{p-1} = 0. \quad (22)$$

We define the initial conditions of the matrix field $M[s_a(t = 0)] = m^a$. As we show in Appendix B, by averaging over the matrix field and implementing the constraints (22), we can re-write (16) in the form:

$$\mathcal{I}(\epsilon_2, q|\epsilon_1) \propto \int_{s_1 \cdot s_2 = Nq} ds_1 ds_2 \int_{s(0)=s_2} \mathcal{D}s^t \mathcal{D}\hat{s}^t \, e^{\mathcal{S}[s^t, \hat{s}^t]} \, \mathcal{K}[s^t, \hat{s}^t], \quad (23)$$

where the proportionality factors do not depend on the dynamical variables $s^t, \hat{s}^t$, and thus do not matter for the derivation of the dynamical equations. In this formula $\mathcal{S}[s^t, \hat{s}^t]$ is the dynamical action, whereas the term $\mathcal{K}[s^t, \hat{s}^t]$ is given by an integral over the initial conditions of the matrix field (19). We give the explicit expressions of these two terms in the following subsection, and refer to the Appendices for the detailed derivations.

### 3.1.3 Order parameters and the role of causality

We report in this subsection the expression of the terms $\mathcal{S}[s^t, \hat{s}^t]$ and $\mathcal{K}[s^t, \hat{s}^t]$ appearing in Eq. (23), that are the relevant ones for the derivation of the dynamical equations given below. The action $\mathcal{S}[s^t, \hat{s}^t]$ can be written as:

$$\mathcal{S}[s^t, \hat{s}^t] = \mathcal{V}_0[s^t, \hat{s}^t] + \mathcal{S}_0[s^t, \hat{s}^t] - \mathcal{S}_B[s^t, \hat{s}^t], \quad (24)$$

where $\mathcal{V}_0[s^t, \hat{s}^t]$ is given in (18). The term $\mathcal{S}_0$ encodes the dynamical evolution given by (6), and it is generic. The term $\mathcal{S}_B$ instead accounts for the peculiar initial conditions of the dynamics: it arises when imposing that the initial condition $s_2$ is a stationary point of energy density $\epsilon_2$, at overlap $q$ from a local minimum of energy $\epsilon_1$. Both actions depend on the dynamical variables only through the two-point functions contained in the $2 \times 2$ matrix $Q(t, t')$ with components:

$$Q(t, t') = \frac{1}{N} \begin{pmatrix} s_2(t) \cdot s_2(t') & s_2(t) \cdot \hat{s}_2(t') \\ s_2(t') \cdot \hat{s}_2(t) & \hat{s}_2(t) \cdot \hat{s}_2(t') \end{pmatrix} \equiv \begin{pmatrix} c(t, t') & r(t, t') \\ r(t', t) & d(t, t') \end{pmatrix}, \quad (25)$$

as well as on the one-point functions contained in the $2 \times 2$ matrices:

$$c(t) = \frac{1}{N} \begin{pmatrix} s_1(0) \cdot s_1(t) & s_1(0) \cdot s_2(t) \\ s_2(0) \cdot s_1(t) & s_2(0) \cdot s_2(t) \end{pmatrix}, \qquad r(t) = \frac{1}{N} \begin{pmatrix} s_1(0) \cdot \hat{s}_1(t) & s_1(0) \cdot \hat{s}_2(t) \\ s_2(0) \cdot \hat{s}_1(t) & s_2(0) \cdot \hat{s}_2(t) \end{pmatrix}. \quad (26)$$

We introduce the vector:

$$\begin{pmatrix} x_1(t) \\ x_2(t) \end{pmatrix} = \begin{pmatrix} c_{12}(t) \\ r_{12}(t) \end{pmatrix}. \quad (27)$$

With this notation, it holds:

$$\mathcal{S}_0[s^t, \hat{s}^t] \to \mathcal{S}_0[Q] = \frac{Np}{4} \int_0^\infty dt \int_0^\infty dt' \{ [c(t, t')]^{p-1} d(t, t') + (p-1)[c(t, t')]^{p-2} r(t, t') r(t', t) \}. \quad (28)$$

This term thus reproduces the dynamical action obtained when starting from random initial conditions [26]. All the non-trivial information on the initial condition of the dynamical evolution is contained in $\mathcal{S}_B$. This term depends explicitly on the overlap $q$, as well as on the energy densities $\epsilon_1, \epsilon_2$. As it appears from the derivation in Appendix D, it contains two contributions:

$$\mathcal{S}_B[s^t, \hat{s}^t] \rightarrow \mathcal{S}_B[Q, x] = \mathcal{S}_B^{(1)}[Q, x] + \mathcal{S}_B^{(2)}[Q, x]. \tag{29}$$

The first contribution $\mathcal{S}_B^{(1)}[Q, x]$ is generated by conditioning $s_a$ to be stationary points. We can write it as:

$$\mathcal{S}_B^{(1)}[Q,x] = \frac{Np}{4} \frac{q^2}{q^2 - q^{2p}} \int_0^\infty dt \int_0^\infty dt' \sum_{a,b=1}^2 \left[\delta_{ab}(1 + q^{p-1}) - q^{p-1}\right]\left[c_{a2}(t)c_{b2}(t')\right]^{p-2} X_{ab}(t,t'), \tag{30}$$

with

$$X_{ab}(t,t') = c_{a2}(t)c_{b2}(t')\left(d(t,t') - \frac{f[r(t), r(t')]}{1-q^2}\right) + r_{a2}(t)r_{b2}(t')\left(c(t,t') - \frac{f[c(t), c(t')]}{1-q^2}\right) +$$
$$+ (p-1)\left(r_{a2}(t)c_{b2}(t') + c_{a2}(t')r_{b2}(t)\right)\left(r(t,t') - \frac{f[c(t), r(t')]}{1-q^2}\right), \tag{31}$$

and where, for arbitrary $2 \times 2$ matrices with components $x_{ab}$ with $a, b \in \{1, 2\}$, we have introduced the form:

$$f(x, y) = x_{12} y_{12} + x_{22} y_{22} - q(x_{12} y_{22} + x_{22} y_{12}). \tag{32}$$

The second contribution $\mathcal{S}_B^{(2)}[Q, x]$ follows from conditioning both on the gradient and on the energy density of the points $s_a$, and reads:

$$\mathcal{S}_B^{(2)}[Q, x] = \frac{1}{2} \frac{p^2(p-1)^2}{4} \sum_{i,j=1}^4 V_i[Q, x] A_{ij} V_j[Q, x], \tag{33}$$

where

$$V_1[Q, x] = p \int_0^\infty dt \, [c_{12}(t)]^{p-1} r_{12}(t) + 2\epsilon_1$$
$$V_2[Q, x] = p \int_0^\infty dt \, [c_{22}(t)]^{p-1} r_{22}(t) + 2\epsilon_2$$
$$V_3[Q, x] = \int_0^\infty dt \, [c_{12}(t)]^{p-1} \frac{r_{22}(t) - q r_{12}(t)}{\sqrt{1-q^2}} + (p-1) \int_0^\infty dt \, [c_{12}(t)]^{p-2} r_{12}(t) \frac{c_{22}(t) - q c_{12}(t)}{\sqrt{1-q^2}}$$
$$V_4[Q, x] = \int_0^\infty dt \, [c_{22}(t)]^{p-1} \frac{r_{12}(t) - q r_{22}(t)}{\sqrt{1-q^2}} + (p-1) \int_0^\infty dt \, [c_{22}(t)]^{p-2} r_{22}(t) \frac{c_{12}(t) - q c_{22}(t)}{\sqrt{1-q^2}}. \tag{34}$$

The matrix $A$ is a $4 \times 4$ matrix given in Appendix D, see Eq. (167) and the following ones.

We now come to the term $\mathcal{K}[s^t, \hat{s}^t]$. This term is obtained as an integral over the $(N-1) \times (N-1)$ matrices $\overline{m}^a$, which denote (up to a shift) the projection of $M[s_a(t=0)] = m^a$ on the tangent plane at $s_a$. Its explicit form reads:

$$\mathcal{K}[s^t, \hat{s}^t] = \int \prod_{a=1}^2 d\overline{m}^a \, e^{-\frac{1}{2}\sum_{\alpha \leq \beta=1}^{N-1} \sum_{\gamma \leq \delta=1}^{N-1} \sum_{a,b=1}^2 \overline{m}_{\alpha\beta}^a [\Omega^*]_{\alpha\beta,\gamma\delta}^{ab} \overline{m}_{\gamma\delta}^b} \prod_{a=1}^2 |\det(\overline{m}^a - \Phi^a[s^t, \hat{s}^t] - p\epsilon_a \mathbb{1})|. \tag{35}$$

From this expression we see that the Hessian matrices $\overline{m}^a$ are Gaussian distributed, with inverse covariances $\Omega^* = [\Sigma^*]^{-1}$ that are given explicitly in Appendix C. The term $\Phi^a[s^t, \hat{s}^t]$ inside the determinant denotes a matrix whose components can be written as:

$$\Phi^a_{\alpha\beta}[s^t, \hat{s}^t] = \phi^a_{\alpha\beta}[s^t, \hat{s}^t] - \delta_{\alpha,N-1}\delta_{\beta,N-1}\mu_a(q, \epsilon_1, \epsilon_2), \tag{36}$$

where we introduced the functions

$$\begin{pmatrix} \mu_1(q, \epsilon_1, \epsilon_2) \\ \mu_2(q, \epsilon_1, \epsilon_2) \end{pmatrix} = \frac{1}{a_2(q)} \begin{pmatrix} \epsilon_2 a_0(q) - \epsilon_1 a_1(q) \\ \epsilon_1 a_0(q) - \epsilon_2 a_1(q) \end{pmatrix}, \tag{37}$$

with:

$$\begin{aligned}
a_0(q) &= p(p-1)(1-q^2)\left[(p-2)q^{2p+2} - (p-1)q^{2p} + q^4\right] \\
a_1(q) &= p(p-1)(1-q^2)q^p\left[q^{2p} - (p-1)q^4 + (p-2)q^2\right] \\
a_2(q) &= q^{6-p} + q^{3p+2} - \left((p-1)^2q^4 - 2(p-2)pq^2 + (p-1)^2\right)q^{p+2}.
\end{aligned} \tag{38}$$

Thus, these matrices are a sum of a rank-1 projector and of a second matrix $\phi^a$ which depends in principle on the dynamical variables $s_2(t), \hat{s}_2(t)$, and can not be expressed compactly in terms of the order parameters (25) and (26). It might therefore seem that the determinants in (35) give a contribution to the action that depends explicitly on the whole time evolution, and that therefore has to be taken into account when deriving the dynamical equations. However, as it appears from the analysis performed in Appendix C, the components of $\phi^a$ vanish when the dynamical average is restricted to trajectories that fulfill the requirement of causality. As a consequence, when the dynamical evolution is causal the matrices $\Phi^a$ reduce to rank-1 projectors, that depend explicitly only on the parameters $q, \epsilon_1$ and $\epsilon_2$ that characterize the initial condition. This is consistent with the natural expectation that the terms appearing in the measure (10), that select the initial condition of the dynamics, are not affected by the subsequent dynamical evolution of the system. Inspecting the distribution of the entries of the matrix $\overline{m}^a$ and the explicit form of the functions $\mu_a(q, \epsilon_1, \epsilon_2)$, one can easily show that the integrand in $\mathcal{K}[s^t, \hat{s}^t]$ reproduces exactly the flat measure over critical points at overlap $q$ with each others, see Appendix C for details. Therefore, accounting for the causality of the dynamical evolution we recover

$$\mathcal{K}[s^t, \hat{s}^t] \xrightarrow{\text{causality}} \mathcal{K}(q, \epsilon_2, \epsilon_1) = \langle \mathcal{N}(\epsilon_1)\mathcal{N}_{s_1}(\epsilon_2, q|\epsilon_1) \rangle, \tag{39}$$

which cancels precisely with the denominator in (15). As it follows from this simplification, all the information on the initial conditions $s_2$ enters in the boundary terms of the dynamical action (24) only. These terms turn out to encode the statistical properties of the Hessian at the initial condition $s_2$, as we show explicitly in Sec. 3.3.1.

## 3.2 Variation of the action and dynamical equations

To finally obtain the dynamical equations, we focus on the relevant term:

$$\int \mathcal{D}s^t \mathcal{D}\hat{s}^t \, e^{\mathcal{V}_0 + \mathcal{S}_0 - \mathcal{S}_B} = \int \mathcal{D}Q \, \mathcal{D}x \, \mathcal{A}[Q, x] \, e^{\mathcal{S}_0[Q] - \mathcal{S}_B[Q, x]}, \tag{40}$$

where we introduced the order parameters (25) and (26), and:

$$\mathcal{A}[Q, x] = \int \mathcal{D}s^t \mathcal{D}\hat{s}^t e^{\mathcal{V}_0[s^t, \hat{s}^t]} \delta\left(NQ_{\alpha\beta}(t, t') - s_2^{(\alpha)}(t) \cdot s_2^{(\beta)}(t')\right) \delta\left(Nx_\alpha(t) - s_1(0)s_2^{(\alpha)}(t)\right), \tag{41}$$

where $s_a^{(1)}(t) = s_a(t)$, $s_a^{(2)}(t) = \hat{s}_a(t)$ and the product over $\alpha, \beta$ is implicit. Using the integral representation of the delta functions, one realizes that the integral over the dynamical variables $s_2(t), \hat{s}_2(t)$ is Gaussian with kernel:

$$M(t, t') = \begin{pmatrix} 0 & (-\partial_t + z(t))\delta(t - t') \\ (\partial_t + z(t))\delta(t - t') & -\alpha\delta(t - t') \end{pmatrix}. \tag{42}$$

Performing the Gaussian integration (see for instance [26]) we obtain:

$$\mathcal{A}[Q, x] = \int \mathcal{D}\Lambda_{\alpha\beta}\, e^{-\frac{N}{2}\overline{a}[\Lambda; Q, x]}, \tag{43}$$

where $\Lambda_{\alpha\beta}(t, t')$ are the auxiliary fields conjugated to $Q_{\alpha\beta}(t, t')$, and the exponent reads:

$$\overline{a}[\Lambda; Q, x] = \operatorname{tr}\{\log(M + 2i\Lambda)\}$$
$$+ \int_0^\infty dt \int_0^\infty dt'\, x(t)(M + 2i\Lambda)(t, t')x(t') - 2i\sum_{\alpha,\beta} \int_0^\infty dt \int_0^\infty dt'\, Q_{\alpha\beta}(t, t')\Lambda_{\alpha\beta}(t, t'). \tag{44}$$

Substituting (43) into (40) and taking the variation with respect to $\Lambda$ and $Q$ we get:

$$M + 2i\Lambda = (Q - xx^T)^{-1}, \qquad \frac{1}{N}\frac{\delta}{\delta Q}[\mathcal{S}_0 - \mathcal{S}_B] + i\Lambda = 0. \tag{45}$$

Combining these two equations and adding the one obtained taking the variation of the action with respect to $x$, we get the coupled equations:

$$M \otimes (Q - xx^T) - \frac{2}{N}\frac{\delta}{\delta Q}[\mathcal{S}_0 - \mathcal{S}_B] \otimes (Q - xx^T) = \mathbb{1},$$
$$M \otimes x = \frac{2}{N}\frac{\delta}{\delta Q}[\mathcal{S}_0 - \mathcal{S}_B] \otimes x + \frac{1}{N}\frac{\delta}{\delta x}[\mathcal{S}_0 - \mathcal{S}_B], \tag{46}$$

where we used the notation $(A \otimes B)_{\alpha\beta}(t, t') = \sum_\gamma \int dt''\, A_{\alpha\gamma}(t, t'')B_{\gamma,\beta}(t'', t')$. A lengthy (but straightforward) calculation of the functional derivatives of the action with respect to the order parameters leads to the dynamical equations reported below (see [35] for details of the derivation in the simplified case in which the boundary terms are absent). We stress that the equations are given under the assumption that the resulting typical dynamical trajectories are causal, meaning that we assume that the saddle-point solution satisfies:

$$d(t, t') = 0 \quad \text{and} \quad r(t, t') = 0 \text{ for } t < t', \tag{47}$$

which implies in particular $r(0, t) = 0$ for any $t > 0$. The remaining equations are for the correlation function $c(t, t')$, the response function $r(t, t')$ for $t > t'$, and the overlap $x(t) \equiv x_1(t) = c_{12}(t)$ with the minimum $s_1$. We report them in the following, and refer to Appendix E for the explicit expression of the constants involved. We make use of the shorthand notation:

$$\gamma_p(q) = \frac{p(p-1)}{2}\frac{q^2}{q^2 - q^{2p}}. \tag{48}$$

### 3.2.1  Dynamical equation for the overlap with the nearby minimum

The equation for $x(t)$ reads:

$$[\partial_t + z(t)]x(t) = \frac{p(p-1)}{2}\int_0^t dt''\, r(t,t'')c^{p-2}(t,t'')x(t'')$$

$$-q\,\gamma_p(q)\int_0^t dt''\, r(t,t'')\left\{c^{p-2}(t)c^{p-1}(t'') - \frac{q^{p-1}}{2}\left(c^{p-2}(t)x^{p-1}(t'') + x^{p-2}(t)c^{p-2}(t'')x(t'')\right)\right\}$$

$$-\gamma_p(q)\int_0^t dt''\, r(t,t'')\left\{x^{p-2}(t)x^{p-1}(t'') - \frac{q^{p-1}}{2}\left(x^{p-2}(t)c^{p-1}(t'') + c^{p-2}(t)x^{p-2}(t'')c(t'')\right)\right\}$$

$$+\mathcal{G}_{\epsilon,q}[c(t),x(t)]\,,$$

$$(49)$$

and $\mathcal{G}_{\epsilon,q}$ depends linearly on the energies, and reads:

$$\mathcal{G}_{\epsilon,q}[c(t),x(t)] = \sum_{a=1}^{2}\epsilon_a\left\{G_1^a(q)c^{p-1}(t) + G_2^a(q)x^{p-1}(t) + G_3^a(q)c^{p-2}(t)x(t) + G_4^a(q)x^{p-2}(t)c(t)\right\},$$

$$(50)$$

and the constants $G_i^a(q)$ are functions of $x(0) = q$, and are reported in Appendix E.

### 3.2.2  Dynamical equations for the correlation function and the Lagrange multiplier

The equation for the correlation $c(t,t')$ reads:

$$[\partial_t + z(t)]c(t,t') = \alpha r(t',t)$$

$$+\frac{p(p-1)}{2}\int_0^t dt''\, r(t,t'')[c(t,t'')]^{p-2}c(t',t'') + \frac{p}{2}\int_0^{t'} dt''[c(t,t'')]^{p-1}r(t',t'')$$

$$-\gamma_p(q)c(t')\int_0^t dt''\, r(t,t'')\left\{c^{p-2}(t)c^{p-1}(t'') - \frac{q^{p-1}}{2}\left(c^{p-2}(t)x^{p-1}(t'') + x^{p-2}(t)x(t'')c^{p-2}(t'')\right)\right\}$$

$$-\gamma_p(q)x(t')\int_0^t dt''\, r(t,t'')\left\{x^{p-2}(t)x^{p-1}(t'') - \frac{q^{p-1}}{2}\left(x^{p-2}(t)c^{p-1}(t'') + c^{p-2}(t)x^{p-2}(t'')c(t'')\right)\right\}$$

$$-\frac{\gamma_p(q)}{p-1}\int_0^{t'} dt''\, r(t',t'')\left\{[x(t)x(t'')]^{p-1} + [c(t)c(t'')]^{p-1} - q^{p-1}\left([x(t)c(t'')]^{p-1} + [c(t)x(t'')]^{p-1}\right)\right\}$$

$$+\mathcal{F}_{\epsilon,q}[c,x]\,,$$

$$(51)$$

where the energy-dependent part is a linear combination of $\epsilon_1,\epsilon_2$ given by:

$$\mathcal{F}_{\epsilon,q}[c,x] = \sum_{a=1}^{2}\epsilon_a\left\{F_1^a(q)\,c^{p-1}(t)c(t') + F_2^a(q)\,c(t')x^{p-1}(t) + F_3^a(q)\,c(t')c^{p-2}(t)x(t) + \right.$$

$$\left. F_4^a(q)\,x(t')x^{p-1}(t) + F_5^a(q)\,x(t')x^{p-2}(t)c(t) + F_6^a(q)\,x(t')c^{p-1}(t)\right\},$$

$$(52)$$

and the constants are given in Appendix E. Setting $t = t'$ we obtain the equation for the multiplier $z(t)$ enforcing the spherical constraint during the dynamics [4]:

$$
\begin{aligned}
z(t) = \frac{\alpha}{2} &+ \frac{p^2}{2} \int_0^t dt'' r(t, t'') [c(t, t'')]^{p-1} \\
&- \gamma_p(q) c(t) \int_0^t dt'' r(t, t'') \left\{ c^{p-2}(t) c^{p-1}(t'') - \frac{q^{p-1}}{2} \left( c^{p-2}(t) x^{p-1}(t'') + x^{p-2}(t) x(t'') c^{p-2}(t'') \right) \right\} \\
&- \gamma_p(q) x(t) \int_0^t dt'' r(t, t'') \left\{ x^{p-2}(t) x^{p-1}(t'') - \frac{q^{p-1}}{2} \left( x^{p-2}(t) c^{p-1}(t'') + c^{p-2}(t) x^{p-2}(t'') c(t'') \right) \right\} \\
&- \frac{\gamma_p(q)}{p-1} \int_0^t dt'' r(t, t'') \left\{ [x(t) x(t'')]^{p-1} + [c(t) c(t'')]^{p-1} - q^{p-1} \left( [x(t) c(t'')]^{p-1} + [c(t) x(t'')]^{p-1} \right) \right\} \\
&+ \mathcal{F}_{\epsilon,q}[c, x] \Big|_{t=t'},
\end{aligned}
$$
(54)

and $\mathcal{F}$ at equal times reduces to:

$$
\mathcal{F}_{\epsilon,q}[c, x] \Big|_{t=t'} = \sum_{a=1}^{2} \epsilon_a \left\{ F_1^a c^p(t) + (F_2^a + F_5^a) x^{p-1}(t) c(t) + (F_3^a + F_6^a) c^{p-1}(t) x(t) + F_4^a x^p(t) \right\}.
$$
(55)

When $t = 0$, setting $x(t=0) = q$ and $c(t=0) = 1$ we get:

$$
z(0) = \frac{\alpha}{2} + \sum_{a=1}^{2} \epsilon_a \left\{ F_1^a + (F_2^a + F_5^a) q^{p-1} + (F_3^a + F_6^a) q + F_4^a q^p \right\} = \frac{\alpha}{2} - p \epsilon_2,
$$
(56)

which for $\alpha = 0$ reduces to the correct value of the Lagrange multiplier enforcing the spherical constraint at a stationary point of energy $N \epsilon_2$.

### 3.2.3 Dynamical equation for the response function

The equation for the response $r(t, t')$ reads:

$$
[\partial_t + z(t)] r(t, t') = \delta(t - t') + \frac{p(p-1)}{2} \int_0^\infty dt'' r(t, t'') r(t'', t') [c(t, t'')]^{p-2}.
$$
(57)

This equation is formally unaltered by the coupling to the initial conditions, the dependence on which is only implicit (through $z(t)$ and $c(t, t')$). This is a generic feature, which occurs also whenever the initial conditions are extracted from a thermal measure [30, 37]. It ultimately follows from the fact that the time evolution of the response function is governed by a memory kernel (the last term in Eq. (57)) whose formal structure depends only on the gradient of the energy functional, and not on the configuration in which the system is initialized.

## 3.3 Two limiting cases: simplifications and checks

We now consider two interesting limits of the above equations, which we remind are derived under the assumption that the initial condition $s(t=0) = s_2$ is a stationary point of energy

---

[4]This equation is obtained starting from the identity:

$$
\left[ \partial_t c(t, t') + \partial_{t'} c(t, t') \right] \Big|_{t, t'=s} = 0.
$$
(53)

In particular, the factor $1/2$ in front of $\alpha$ comes from the fact that only one of these two derivatives gives a non-zero contribution multiplying $\alpha$, while all the other terms are doubled.

density $\epsilon_2$, at overlap $q$ with a local minimum $s_1$ of energy density $\epsilon_1$.

The first case we focus on consists in the limit $\alpha = 2T \to 0$, when the noise in the Langevin equation vanishes and the dynamics reduces to gradient descent starting from a stationary point $s_2$. As we shall see, and as expected, if this point $s_2$ is a minimum then the system remains stuck there, otherwise if this point is a saddle a dynamical instability takes place.

The second case corresponds to the limit $q \to 0$, when the initial condition decouples from $s_1$, and one samples uniformly all stationary points at a given energy. For this reason we will refer to it as "microcanonical initial conditions". This limit is useful to check our equations since it can be connected to the one analyzed in [37].

### 3.3.1 Gradient descent from a stationary point: the "static" solution and its stability

In the noiseless limit $\alpha \to 0$, the dynamical equations must admit a solution in which the system does not move away from the initial condition, given that the latter is a stationary point. We refer to this as the "static" solution. It is easy to check using the explicit form of the constants given in Appendix E that $x(t) = q$ and $c(t, t') = 1$ solve the above equations in this limit. Indeed, plugging this ansatz into (49) we get:

$$z(t)q = z_0 q = \mathcal{G}_{\mathcal{E},q}[1, q] = \sum_{a=1}^{2} \epsilon_a \left\{ G_1^a(q) + G_2^a(q)q^{p-1} + G_3^a(q)q + G_4^a(q)q^{p-2} \right\} = -pq\epsilon_2, \tag{58}$$

which rightly gives the value of the zero-time multiplier $z(0) = z_0 = -p\epsilon_2$. The same identity is obtained from (51). The equation (57) for the response becomes:

$$(\partial_t + z_0) r(t, t') = \delta(t - t') + \frac{p(p-1)}{2} \int_0^\infty dt'' r(t, t'') r(t'', t'). \tag{59}$$

Assuming time-translation invariance, this is equivalent to

$$(\partial_\tau + z_0) R(\tau) = \delta(\tau) + \frac{p(p-1)}{2} \int_0^\tau d\tau' R(\tau - \tau') R(\tau'). \tag{60}$$

The Laplace transform of this equation is simply:

$$[\omega + z_0] \hat{R}(\omega) = 1 + \frac{p(p-1)}{2} \left[ \hat{R}(\omega) \right]^2, \tag{61}$$

where we used that the Laplace transform of the derivative is $\omega \hat{R}(\omega) - R(0^-)$, and $R(0^-) = 0$. The equation admits the shifted GOE resolvent as a solution, i.e.,

$$\hat{R}(\omega) = G_{\overline{\sigma}}(\omega + z_0), \qquad z_0 = -p\epsilon_2, \qquad \overline{\sigma}^2 = \frac{\sigma^2}{2} = \frac{p(p-1)}{2}, \tag{62}$$

where the function $G$ is given in (116). The inverse Laplace transform is proportional to a Bessel function,

$$R(\tau) = \frac{e^{-z_0 \tau}}{\overline{\sigma} \tau} I_1(2\overline{\sigma}\tau). \tag{63}$$

This result coincides with the one of stochastic dynamics in purely quadratic landscapes [40, 41], as in the noiseless limit the non-linear part of the potential is not explored.

The initial condition $s_2$ has a Hessian whose statistics depends on the parameters $q, \epsilon_1$ and $\epsilon_2$, as recalled in Appendix A. Its eigenvalue density is almost entirely positive definite (and GOE-like), with the exception of possibly one negative eigenvalue that appears for certain values of the parameters (given by the condition (114)). When the initial condition $s_2$ is a saddle

with one single negative eigenvalue, the "static" solution must be dynamically unstable, since there exist a direction in configuration space in which the landscape has negative curvature, allowing the system to escape from the stationary point, see Fig. 1. For fixed $q$, this happens whenever the initial condition $s_2$ is chosen to have energy $\epsilon_2 \in [\epsilon_{\text{ms}}, \overline{\epsilon}]$, see Fig. 2. In order to check this instability from the dynamical equations, we consider the linearization of Eq. (49) around the static solution $x(t) = q$. Setting $x(t) = q + \delta x(t)$, we get:

$$\frac{d}{dt}\delta x(t) = \mathcal{O}[\delta x](t), \tag{64}$$

where the operator $\mathcal{O}$ acts as:

$$\mathcal{O}[\delta x](t) = \left(-z_0 + \tilde{\mathcal{G}}_{\epsilon,q} - \frac{p(p-1)}{2}\frac{p-2}{2}\frac{q^{2p-2}(1-q^2)}{q^2-q^{2p}}\int_0^t dt'' R(t-t'')\right)\delta x(t)$$
$$+ \frac{p(p-1)}{2}\left(1 - \frac{p}{2}\frac{q^{2p-2}(1-q^2)}{q^2-q^{2p}}\right)\int_0^t dt'' R(t-t'')\delta x(t''), \tag{65}$$

and $R(\cdot)$ is the response in the stationary point with energy density $\epsilon_2$, $z_0 = -p\epsilon_2$ and $\tilde{\mathcal{G}}_{\epsilon,q}$ reads:

$$\tilde{\mathcal{G}}_{\epsilon,q} = \sum_{a=1}^2 \epsilon_a \left[G_2^a(p-1)q^{p-2} + G_3^a + G_4^a(p-2)q^{p-3}\right] = \frac{\epsilon_2 a_1(q) - \epsilon_1 a_0(q)}{a_2(q)}, \tag{66}$$

with the $a_i(q)$ given in (38). The static solution becomes unstable when the linear operator $\mathcal{O}$ has eigenvalues that becomes positive. We assume that $\delta x(s)$ is slowly varying, which is correct close to the transition where the instability is small. As a consequence, we can extract it from the integration in (65). Taking $t \to \infty$ we get:

$$\mathcal{O}[\delta x](t_\infty) = \lambda_\infty \delta x(t_\infty), \tag{67}$$

with

$$\lambda_\infty = -z_0 + \tilde{\mathcal{G}}_{\epsilon,q} + \frac{p(p-1)}{2}\left[1 - \frac{(p-1)q^{2p-2}(1-q^2)}{q^2-q^{2p}}\right]\int_0^\infty dt'' R(t''). \tag{68}$$

Using that the integral is the Laplace transform evaluated at zero, which is related to the GOE resolvent $G_{\overline{\sigma}}$ with $\overline{\sigma}^2 = p(p-1)/2$ as:

$$\int_0^\infty dt'' R(t'') = \hat{R}(0) = G_{\overline{\sigma}}(z_0), \tag{69}$$

see Eq. (62), one finds that

$$\lambda_\infty = -z_0 + \tilde{\mathcal{G}}_{\epsilon,q} + \frac{p(p-1)}{2}\left[1 - \frac{(p-1)q^{2p-2}(1-q^2)}{q^2-q^{2p}}\right]G_{\overline{\sigma}}(z_0). \tag{70}$$

Using (66) we get that the instability condition $\lambda_\infty = 0$ reads:

$$p\epsilon_2 + \frac{\epsilon_2 a_1(q) - \epsilon_1 a_0(q)}{a_2(q)} + p(p-1)\left[1 - \frac{(p-1)q^{2p-2}(1-q^2)}{q^2-q^{2p}}\right]\frac{G_{\overline{\sigma}}(-p\epsilon_2)}{\sqrt{2}} = 0. \tag{71}$$

This equation is precisely equivalent to the one corresponding to the isolated eigenvalue of the Hessian at $s_2$ being equal to zero. Indeed, as we recall in Appendix A the isolated eigenvalue of the Hessian is given by:

$$\lambda_0(q, \epsilon_1, \epsilon_2) = \lambda_{\text{min}}(q, \epsilon_1, \epsilon_2) - \sqrt{2}p\epsilon_2, \tag{72}$$

where $\lambda_{\min}$ solves the equation

$$\lambda - \mu(q, \epsilon, \epsilon_0) - \Delta^2(q)G_\sigma(\lambda) = 0, \tag{73}$$

with

$$\mu(q, \epsilon, \epsilon_0) = -\frac{\sqrt{2}\epsilon_2 a_1(q) - \sqrt{2}\epsilon_1 a_0(q)}{a_2(q)}, \qquad \Delta^2(q) \equiv p(p-1)\left[1 - \frac{(p-1)q^{2p-2}(1-q^2)}{q^2 - q^{2p}}\right]. \tag{74}$$

Multiplying (71) by $\sqrt{2}$ and using that $G_{\frac{\sigma}{\sqrt{2}}}(z) = -\sqrt{2}G_\sigma\left(-\sqrt{2}z\right)$ we obtain

$$\sqrt{2}p\epsilon_2 - \mu(q, \epsilon_1, \epsilon_2) - \Delta^2(q)G_{\overline{\sigma}}(\sqrt{2}p\epsilon_2) = 0, \tag{75}$$

which corresponds to $\lambda_{\min} = \sqrt{2}p\epsilon_2$ and thus $\lambda_0(q, \epsilon_1, \epsilon_2) = 0$. Therefore, the dynamical solution $c(t, t') = 1$ and $x(t) = q$ becomes unstable exactly at the values of parameters at which $s_2$ undergoes a transition from being a minimum to being a saddle, as expected.

### 3.3.2 Microcanonical initial conditions

We now consider the case in which $q \to 0$, where the initial condition of the dynamics $s_2$ decorrelates from the minimum $s_1$. In this limit, the only non-vanishing $G_k^a(q)$ constant is $G_2^1(q) \to -p$, while the non-vanishing $F_k^a(q)$ constants are $F_1^2(q), F_4^1(q) \to -p$. The equation for $x(t)$ reduces to:

$$[\partial_t + z(t)]x(t) = \frac{p(p-1)}{2}\int_0^t dt'' r(t, t'')\left[c^{p-2}(t, t'')x(t'') - x^{p-2}(t)x^{p-1}(t'')\right] - p\,\epsilon_1 x^{p-1}(t), \tag{76}$$

which is homogeneous and thus admits the solution $x(t) \equiv 0$ for $x(0) = q = 0$. The equation for the correlation when $x(t) = 0$ for any $t$ reduces to:

$$[\partial_t + z(t)]c(t, t') = \alpha r(t', t) + \frac{p(p-1)}{2}\int_0^t dt'' r(t, t'')[c(t, t'')]^{p-2}c(t', t'')$$
$$+ \frac{p}{2}\int_0^{t'} dt''[c(t, t'')]^{p-1}r(t', t'') - \frac{p(p-1)}{2}c(t')\int_0^t dt'' r(t, t'')c^{p-2}(t)c^{p-1}(t'') \tag{77}$$
$$- \frac{p}{2}\int_0^{t'} dt'' r(t', t'')[c(t)c(t'')]^{p-1} - p\epsilon_2 c^{p-1}(t)c(t'),$$

while the Lagrange multiplier reads:

$$z(t) = \frac{\alpha}{2} - p\epsilon_2 c^p(t) + \frac{p^2}{2}\int_0^t dt'' r(t, t'')c^{p-1}(t, t'') - \frac{p^2}{2}c^{p-1}(t)\int_0^t dt'' c^{p-1}(t'')r(t, t''). \tag{78}$$

These equations give the evolution of the correlation function for a dynamics conditioned to start from a typical stationary point of energy density $\epsilon_2$, which therefore will be a local minimum for $\epsilon_2 < \epsilon_{\text{th}}$. The first two terms in the second line of (77) and the last term in (78) are generated by conditioning on the stationarity of the initial condition: setting them to zero, we get the dynamical equations conditioned to start from a point extracted with uniform measure from the manifold at a given energy density $\epsilon_2$.

This is a case that has been already considered in the literature: it is the microcanonical equivalent of the one analyzed in [37], where the initial condition is extracted with a Boltzmann

measure at a temperature $T'$ between the static and the dynamical transition temperatures. It provides a useful check of our method, which is different from the one followed in [37]. In fact, we recover the same dynamical equations, in particular the same boundary terms [5].

# 4 Where does the system fall when escaping from the saddle?

The aim of this section is to study where the system falls after escaping from the saddle illustrated in Fig.1. One (trivial) possibility is to come back to the original reference minimum. The other possibility—the interesting one—is that the system lands in a different basin. In order to analyze this case, we consider the large time limit in which the system equilibrates within the basin. This allows us to obtain closed equations describing the properties of the basin, or more precisely the minimum since we consider the small-noise case.

## 4.1 Equations for the minimum: asymptotic analysis of the dynamics after the fall from the saddle

When the initial condition $s_2$ is an unstable saddle, in presence of weak thermal fluctuations ($\alpha = 2T \neq 0$) the system eventually escapes from it (even though this might require extremely large times). In this section we study the asymptotic solutions of the dynamical equations representing the dynamics within the basin that has been reached after escaping from the saddle. We therefore assume that after a finite time $t_{eq}$ a stationary limit is reached (see [38] for a similar computation), meaning that the one-point functions converge to a time-independent value:

$$x(t) \overset{t\to\infty}{\longrightarrow} q_{13}, \quad c(t) \overset{t\to\infty}{\longrightarrow} q_{23}, \quad z(t) \overset{t\to\infty}{\longrightarrow} z_3\,, \tag{80}$$

and the two point functions become time translation invariant:

$$r(t,t') \overset{t,t'\to\infty}{\longrightarrow} R(t-t'), \quad c(t,t') \overset{t,t'\to\infty}{\longrightarrow} C(t-t')\,. \tag{81}$$

Moreover, we assume that in the asymptotic limit the dynamics equilibrates into some local minimum of the energy landscape, and that the fluctuation-dissipation relation holds at large times:

$$R(\tau) = -\beta\, \partial_\tau C(\tau), \quad \text{with} \quad C(\tau) \overset{\tau\to\infty}{\longrightarrow} A_3\,. \tag{82}$$

Here $\beta$ is the inverse temperature. In the limit of zero temperature, if the dynamics ends up asymptotically in a minimum then $C(\tau) \to 1$ (and notice that $C(0) = 1$). To capture the dynamical evolution it is necessary to introduce the scaling variable $\phi(\tau) = \beta(1 - C(\tau))$, with stationary value $\phi_3 = \beta(1 - A_3)$. Assuming that the initial transient decouples from the long-time dynamics:

$$\lim_{t\to\infty} \int_0^{t_{eq}} dt''R(t,t'')\{\cdots\} = 0\,, \tag{83}$$

the equation for $z(t)$ becomes for $t, t' \to \infty$:

$$z_3 = \frac{\alpha}{2} + \frac{p\beta}{2}(1-A_3^p) + \tilde{F}\,, \tag{84}$$

---

[5]For a comparison, one needs to keep in mind that the Lagrange multiplier $\mu(t)$ in [37] and the $z(t)$ in this work are related by

$$z(t) = \mu(t) + \frac{p(p-1)}{2}\int_0^t dt''r(t,t'')c^{p-2}(t,t'') + \frac{p}{2T'}c^{p-1}(t,0)\,. \tag{79}$$

with:

$$
\tilde{F} = -\frac{p^2}{2}\frac{q^2}{q^2 - q^{2p}}\beta(1 - A_3)\Big[ q_{23}^{2p-2} - 2q^{p-1}q_{23}^{p-1}q_{13}^{p-1} + q_{13}^{2p-2}\Big]
$$
$$
+ \sum_{a=1}^{2}\epsilon_a\Big[ F_1^a q_{23}^p + (F_2^a + F_5^a)q_{13}^{p-1}q_{23} + (F_3^a + F_6^a)q_{23}^{p-1}q_{13} + F_4^a q_{13}^p\Big],
$$
(85)

and $\beta(1 - A_3^p) \approx p\beta(1 - A_3)$. The equation for the correlation with these assumptions is

$$
\partial_t C(t - t') + z_3 C(t - t') = \frac{p\beta}{2}(C(t - t') - A_3^p) - \frac{p\beta}{2}\int_{t'}^{t}dt''\, C^{p-1}(t - t'')\partial_{t''}C(t'' - t') + \tilde{F},
$$
(86)

and using (85) we get:

$$
\partial_\tau C(\tau) - z_3(1 - C(\tau)) = -\frac{\alpha}{2} - \frac{p\beta}{2}(1 - C(\tau)) - \frac{p\beta}{2}\int_0^\tau d\tau''\, C^{p-1}(\tau'')\partial_{\tau''}C(\tau - \tau''). \quad (87)
$$

Setting $\phi(\tau) = \beta(1 - C(\tau))$ and

$$
C^{p-1}(\tau) \approx 1 - (p - 1)\phi(\tau)\beta^{-1}, \qquad \beta\partial_\tau C(\tau) = -\partial_\tau\phi(\tau), \qquad (88)
$$

we finally obtain:

$$
\partial_\tau\phi(\tau) + z_3\phi(\tau) = \frac{\alpha}{2T} + \frac{p(p-1)}{2}\int_0^\tau d\tau'\,\phi(\tau - \tau')\partial_{\tau'}\phi(\tau'), \qquad (89)
$$

which has a finite limit when $T \to 0$ because the correlation of the noise is $\alpha = 2T$; integrating the equation for the response from $t'$ to $t$ reproduces (87). In the limit $\tau \to \infty$ we get:

$$
z_3\phi_3 = 1 + \frac{p(p-1)}{2}\phi_3^2, \qquad (90)
$$

using that $\phi(0) = 0$. The fluctuation-dissipation relation implies that $\phi_3$ coincides with the static susceptibility in the minimum reached asymptotically by the dynamics. The equation above is indeed consistent with this interpretation since $\phi_3$ satisfies the same equation of $G_{\overline{\sigma}}(z_3)$, where $G_{\overline{\sigma}}$ is the GOE resolvent which is directly related to the static susceptibility, see Eq. (116), as well as (62).

Additional relations between the parameters $q_{13}, q_{23}, z_3$ and $\phi_3$ are obtained from the $t \to \infty$ limit of the equations for $x(t), z(t)$ and $c(t)$, which gives the following coupled equations:

$$
z_3 q_{23} = \frac{p(p-1)}{2}\phi_3\Big[ q_{23} - \frac{q^2}{q^2 - q^{2p}}\Big( q_{23}^{2p-3} - q^{p-1}q_{13}^{p-1}q_{23}^{p-2} - q^p q_{23}^{p-1}q_{13}^{p-2} + qq_{13}^{2p-3}\Big)\Big]
$$
$$
+ \sum_{a=1}^{2}\epsilon_a\Big[ q_{23}^{p-1}(F_1^a + qF_6^a) + q_{13}^{p-1}(F_2^a + qF_4^a) + q_{23}^{p-2}q_{13}F_3^a + qF_5^a q_{13}^{p-2}q_{23}\Big],
$$
(91)

and

$$
z_3 = \frac{p^2}{2}\phi_3 - \frac{p^2}{2}\frac{q^2}{q^2 - q^{2p}}\phi_3\Big( q_{23}^{2p-2} - 2q^{p-1}q_{13}^{p-1}q_{23}^{p-1} + q_{13}^{2p-2}\Big)
$$
$$
+ \sum_{a=1}^{2}\epsilon_a\Big[ F_1^a q_{23}^p + q_{13}^{p-1}q_{23}(F_2^a + F_5^a) + q_{23}^{p-1}q_{13}(F_3^a + F_6^a) + F_4^a q_{13}^p\Big],
$$
(92)

and

$$
\begin{aligned}
z_3 q_{13} &= \frac{p(p-1)}{2} \phi_3 \left[ q_{13} - \frac{q^2}{q^2 - q^{2p}} \left( q_{13}^{2p-3} - q^{p-1} q_{13}^{p-2} q_{23}^{p-1} - q^p q_{23}^{p-2} q_{13}^{p-1} + q q_{23}^{2p-3} \right) \right] \\
&\quad + \sum_{a=1}^{2} \epsilon_a \left[ G_1^a q_{23}^{p-1} + G_2^a q_{13}^{p-1} + G_3^a q_{23}^{p-2} q_{13} + G_4^a q_{13}^{p-2} q_{23} \right].
\end{aligned}
\tag{93}
$$

The solution of these four coupled equations gives information on the minima reached asymptotically by the dynamics, as a function of the parameters $q$ and $\epsilon_1, \epsilon_2$ that specify the initial conditions. In particular, the energy of the minimum $s_3$ reached asymptotically by the dynamics can be read out from $z_3$.

As we anticipated, we expect two kinds of solutions for these equation when the initial condition $s_2$ is an unstable saddle. One solution should correspond to the trajectory that escapes from the saddle and goes back to the original minimum $s_1$. In fact, we do find that this set of equations admits the solution $q_{13} = 1$ and $q_{23} = q$, and substituting these values into (91), (92) and (93) we obtain the identity $z_3 = -p\epsilon_1$. This indicates that the two stationary points $s_1$ and $s_2$ are not only geometrically connected (meaning that the unstable direction of the saddle $s_2$ is oriented towards the minimum $s_1$ in configuration space) but also *dynamically* connected, since there exists a solution of the dynamical equations that corresponds to the relaxation from the saddle to the reference minimum, see also Sec. 5.

The other (less-trivial) solution of the above system of equations instead corresponds to the system relaxing to another local minimum $s_3$ that is connected to the reference one through the index-1 saddle $s_2$. We focus on this second solution in the following.

## 4.2 Geometrical arrangement of the minimum past the barrier and the saddle in configuration space

We now want to discuss the correlations between the pairs of minima connected by the index-1 saddles. In order to do so, we solve the asymptotic Eqs. (91), (92) and (93) for the parameters $q_{23}, q_{13}$ and $z_3$. Given $z_3$, the response $\phi_3$ is then readily obtained solving the quadratic equation (90). We choose a representative energy of the reference minimum, equal to $\epsilon_1 = -1.167$ as in Fig. 2 (recall that $\epsilon_{gs} \approx -1.172$ and $\epsilon_{th} \approx -1.1547$). From the study of the constrained complexity [22], we know that index-1 saddles are the dominant stationary points in a range of energies and overlaps corresponding to the violet region in the figure: for any $\epsilon_2 \in \left[ \epsilon_2^*, \epsilon_{th} \right]$ we find that the typical stationary points are saddles if $q \in [q_{ms}(\epsilon_2), q_m(\epsilon_2)]$. For the chosen $\epsilon_1$, the deepest energy of these saddles is $\epsilon_2^* \approx -1.158$, and the corresponding stationary points are found at $q^* \approx 0.677$; the range of allowed overlaps is maximal for the saddles that are at the threshold energy, where $q_m(\epsilon_{th}) = q_M \approx 0.757$. Beyond this value of the overlap, the landscape is typically devoid of stationary points (besides the reference minimum).

As shown in Sec. 3.3.1, in this regime of $\epsilon_2, q$ the static solution of the dynamical equations (corresponding to $q_{23} = 1$ and $q_{13} = q$) is unstable, and another solution of the asymptotic equations is found, with $q_{23} < 1$. We denote with $\epsilon_3$ the energy density of the minimum that is reached asymptotically by the dynamics. Fig. 3 shows the values of this asymptotic energy density as a function of the energy of the saddle $\epsilon_2$ and of its overlap $q$ with the reference minimum, as well as the values of the asymptotic overlaps $q_{13}$ with the reference minimum. The following features are observed:

- *At fixed energy $\epsilon_2$ of the saddle*, the asymptotic energy $\epsilon_3$ decreases with $q$, meaning that the saddles that are closer to the reference minimum connect the latter to minima that lie deeper in the landscape; the same holds true for the overlap $q_{13}$ for sufficiently small values of $\epsilon_2$ (see the caption of Fig. 5 for more details). Therefore, among the

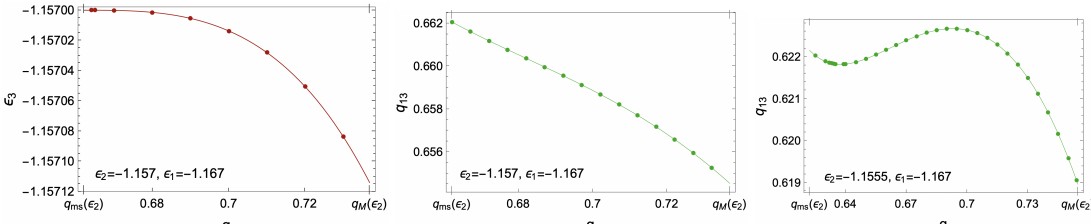

Figure 5: *Left.* The energy $\epsilon_3$ of the minima reached asymptotically by the dynamics decreases with $q$, at fixed energy $\epsilon_2$ of the saddle. *Middle and Right.* The behavior of the asymptotic overlap $q_{13}$ with $q$ depends on $\epsilon_2$: for $\epsilon_2$ sufficiently small, $q_{13}$ decreases with $q$, i.e., the closer is the saddle to the reference minimum, the farther is the one reached asymptotically; for energies $\epsilon_2$ closer to the threshold, the behavior of $q_{13}$ is non-monotonic.

saddles at the same depth in the landscape, the ones that are closer to the minima lead to a more efficient exploration of configuration space, as they allow to explore farther regions and to reach deeper minima. We recall that increasing $q$ corresponds to selecting saddles that are less numerous (have lower complexity) and that are in general steeper along the direction connecting to the reference minimum (as they have a smaller isolated eigenvalue).

- *At fixed overlap $q$* with the reference minimum, deeper saddles connect the latter with local minima with smaller energy. In particular, the deepest minimum that can be reached through this family of index-1 saddles is connected to the reference one through the lowest saddle of energy $\epsilon_2^*$. However, this is not the farthest point that can be reached through this family of index-1 saddles.

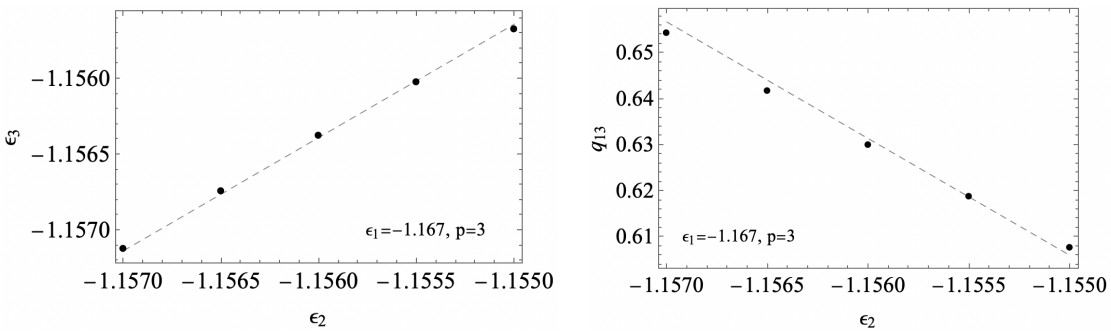

Figure 6: Asymptotic overlap (*Left.*) and energy (*Right.*) of the minima $s_3$ reached from the zero-complexity saddles at energy $\epsilon_2$ that are closer to the reference minimum (*i.e.*, that are at overlap $q_M(\epsilon_2)$). The dashed line are linear fits.

In Fig. 6 we focus on the closest saddles to the minimum for each $\epsilon_2$ (i.e., on those at overlap $q_M(\epsilon_2)$ with the minimum, having zero complexity), and plot the asymptotic energy and overlaps reached from these saddles, which show an almost linear dependence on $\epsilon_2$. We see that moving along the curve corresponding to zero complexity of the saddles, the ones having lower energy lead to lower energy minima, but that are at larger overlap with the original minimum. Thus, there is a competition between energy and overlap of the asymptotic states: the saddles leading to lowest energies are not those leading to the farthest stationary points.

More generally, the asymptotic analysis shows that the minima that are reached through this family of saddles have a distribution in energy concentrated around values that are much

higher than $\epsilon_1$ (the energy of the reference minimum), and are rather close to the energy $\epsilon_2$ of the saddles. Moreover, the asymptotic correlation with the initial condition (the saddle) remains quite close to one, as we show in Fig. 7. This suggest that the minima reached asymptotically are close to the saddles in configuration space. Moreover, we find that they are correlated to the reference minimum [6]. Indeed, the corresponding parameters $(q_{13}, \epsilon_3)$ lie in a region of configuration space that is dominated by minima having an Hessian that feels the presence of the reference minimum through a single (positive) isolated eigenvalue, see Fig. 7 and the comparison with Fig. 2.

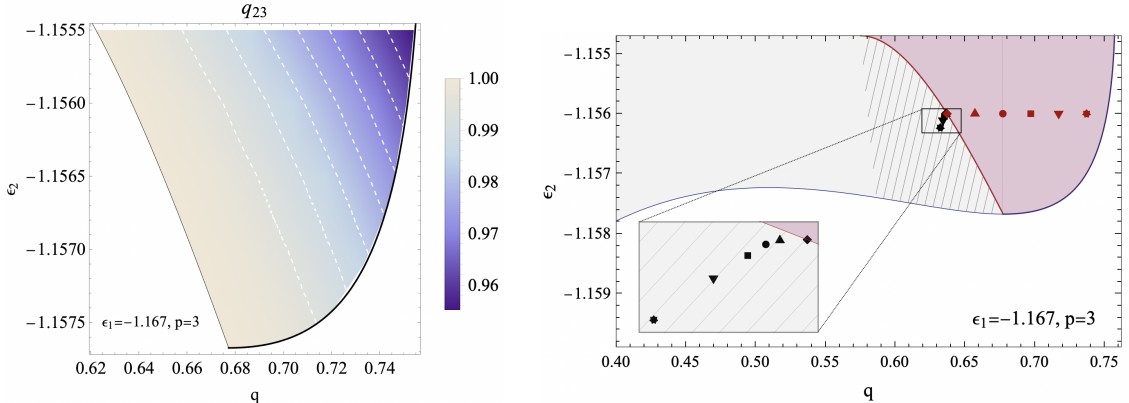

Figure 7: *Left.* Asymptotic correlation function $q_{23}$, giving the overlap between the minimum $s_3$ reached asymptotically by the dynamics and the saddle $s_2$ chosen as initial condition. The white dashed lines are level curves. The flatter is the negative direction of the saddle (*i.e.*, the closer is $\epsilon_2$ to $\epsilon_{ms}$), the closer is the minimum reached asymptotically. *Right.* The red points represent the parameters of the saddles chosen as initial conditions for the dynamics, while the black ones are the parameters of the minima reached asymptotically from the saddles with the same symbol. The inset is a zoom of these points. The saddles that have a flatter unstable direction (those at smaller $q$) lead to closer local minima. All minima reached asymptotically lie in the region of configuration space that is dominated by minima correlated to the reference one, having one positive isolated eigenvalue (dashed gray area).

## 5 Numerical solution and free-fall dynamics

The purpose of this section is to present a full numerical solution of the equations (49), (51), (54), (57). We shall show that after escaping from the selected saddle the system displays a relaxation dynamics towards the connected minima, thus validating the assumptions behind the asymptotic solution obtained in Sec. 4 (in particular we exclude the existence of aging dynamics and trapping in spurious minima). The numerical solution of the free-fall dynamics from the saddle will be instrumental in reconstructing the shape of the dynamical instanton in the next section, see Fig. 8

---

[6]In this discussion we restrict to initial conditions lying in a region of configuration space where the complexity of stationary points is non-negative, *i.e.*, to $q < q_M(\epsilon_2)$. For $q > q_M(\epsilon_2)$, non-trivial solutions of the asymptotic dynamical equations can still be found; however, for $q$ large enough, they lie in a region of configuration space where stationary points of energy $\epsilon_3$ are exponentially rare (their complexity is negative).

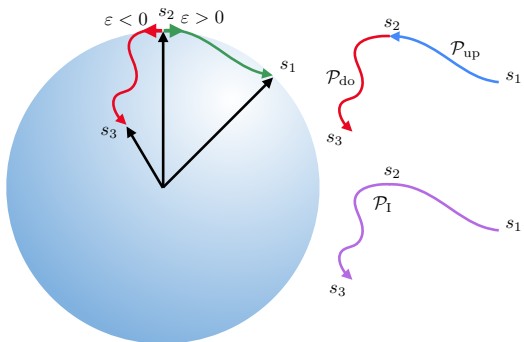

Figure 8: Schematic representation of steps for numerical integration and instanton reconstruction. Kicks with amplitudes of opposite signs allow numerical integration of dynamical paths from the saddle towards the original minimum and from the saddle away from the original minimum. The second path is $\mathcal{P}_{\mathrm{do}}$. The time reversal of the first path is $\mathcal{P}_{\mathrm{up}}$. The dynamical instanton path $\mathcal{P}_{\mathrm{I}}$ is obtained by joining $\mathcal{P}_{\mathrm{up}}$ and $\mathcal{P}_{\mathrm{do}}$.

## 5.1 Kicking the system out of the saddle

As already discussed in the previous section, when the initial condition is on the saddle the system remains stuck there even though this is an unstable point. The reason is that this unique unstable direction is one out of $N$, so in the large $N$ limit the system does not escape from the saddle in any finite time. By linearizing the dynamics around the unstable saddle is easy to establish that the escape time equals $\ln(N/\alpha)/|\lambda_0|$, i.e. it increases logarithmically with $N$ ($\lambda_0$ is the negative eigenvalue of the Hessian corresponding to the unstable direction). In the following, since we are interested in the free-fall dynamics, we bypass this slow process by introducing a small perturbation aligned, or counter-aligned, with the unique unstable direction of the saddle. We implement this perturbation in the form of an impulse, a kick, of infinitesimal amplitude and duration in the direction of $s_1$ which has a finite projection on the unstable direction [22, 23], i.e. along the vector $s_1 - s_2$. However since the component along $s_2$ is compensated anyway by the spherical constraint we simplify and consider a kick in the direction $s_1 - q s_2$ (see Fig. 8) perpendicular to $s_2$. This leads to the modified dynamical equations:

$$\partial_t s_i(t) = -\frac{\delta \mathcal{E}[s^t]}{\delta s_i(t)} - z(t) s_i(t) + \xi_i(t) + \varepsilon \delta(t)[s_1 - q s_2]_i, \tag{94}$$

with initial condition $s(t=0) = s_2$ chosen as usual. For $\varepsilon > 0$ ($< 0$), the kick pushes the system towards (away from) the minimum $s_1$. In the second case the convergence to the other minimum $s_3$ is favored. The equations for $x(t)$, $c(t,t')$ and $z(t)$ change in a very simple way that can be read from (94) and only affects the contributions coming from the initial condition $\mathcal{G}_{\epsilon,q}[c,x]$ and $\mathcal{F}_{\epsilon,q}[c,x]$ in the following way:

$$\begin{aligned}
\mathcal{G}_{\epsilon,q}^{\varepsilon}[c,x] &= \mathcal{G}_{\epsilon,q}[c,x] + \varepsilon \delta(t)[1-q^2] \\
\mathcal{F}_{\epsilon,q}^{\varepsilon}[c,x] &= \mathcal{F}_{\epsilon,q}[c,x] + \varepsilon \delta(t)[x(t') - q c(t')] .
\end{aligned} \tag{95}$$

The equation for $r(t,t')$ that is not explicitly affected by initial conditions would change uniquely through the Lagrange multiplier $z(t)$, which has itself a null contribution $\delta(t)[x(t) - q c(t)] = 0$ from this kick by construction. The simplest form for the new sys-

tem equations can then be rewritten using (51), (54) and (48) as:

$$[\partial_t + z(t)]c(t,t') = \alpha r(t',t) + \frac{p(p-1)}{2}\int_0^t dt'' r(t,t'')[c(t,t'')]^{p-2}c(t',t'')$$

$$+\frac{p}{2}\int_0^{t'} dt''[c(t,t'')]^{p-1}r(t',t'')$$

$$-\gamma_p(q)c(t')\int_0^t dt'' r(t,t'')\left\{c^{p-2}(t)c^{p-1}(t'')-\frac{q^{p-1}}{2}\left(c^{p-2}(t)x^{p-1}(t'')+x^{p-2}(t)x(t'')c^{p-2}(t'')\right)\right\}$$

$$-\gamma_p(q)x(t')\int_0^t dt'' r(t,t'')\left\{x^{p-2}(t)x^{p-1}(t'')-\frac{q^{p-1}}{2}\left(x^{p-2}(t)c^{p-1}(t'')+c^{p-2}(t)x^{p-2}(t'')c(t'')\right)\right\}$$

$$-\frac{\gamma_p(q)}{p-1}\int_0^{t'} dt'' r(t',t'')\left\{[x(t)x(t'')]^{p-1}+[c(t)c(t'')]^{p-1}-q^{p-1}\left([x(t)c(t'')]^{p-1}+[c(t)x(t'')]^{p-1}\right)\right\}$$

$$+\mathcal{F}_{\epsilon,q}[c,x]+\varepsilon\delta(t)[x(t')-qc(t')]\,,$$
(96)

and

$$[\partial_t + z(t)]x(t) = \frac{p(p-1)}{2}\int_0^t dt'' r(t,t'')c^{p-2}(t,t'')x(t'')$$

$$-\gamma_p(q)q\int_0^t dt'' r(t,t'')\left\{c^{p-2}(t)c^{p-1}(t'')-\frac{q^{p-1}}{2}\left(c^{p-2}(t)x^{p-1}(t'')+x^{p-2}(t)c^{p-2}(t'')x(t'')\right)\right\}$$

$$-\gamma_p(q)\int_0^t dt'' r(t,t'')\left\{x^{p-2}(t)x^{p-1}(t'')-\frac{q^{p-1}}{2}\left(x^{p-2}(t)c^{p-1}(t'')+c^{p-2}(t)x^{p-2}(t'')c(t'')\right)\right\}$$

$$+\mathcal{G}_{\epsilon,q}[c,x]+\varepsilon\delta(t)[1-q^2]\,.$$
(97)

From the last new equation it becomes evident that, if for $\varepsilon = 0$ $x(t) = q$ $\forall t$, setting $\varepsilon > 0$ ($< 0$) leads to an initial increase (decrease) of $x(t)$ from $q$ and therefore a consequent relaxation towards (away from) $s_1$, as pictorially represented in Fig. 8.

## 5.2 Numerical integration scheme

The algorithm used to integrate the dynamical equations is a modification of the code developed for the Cugliandolo-Kurchan equations on a fixed time-grid used in [42,43] and available at https://github.com/sphinxteam/spiked_matrix-tensor (see also [44, 45] for early works on the numerical integration of similar equations).

We introduced two modifications to it. The first one consists in adding the terms of the equations derived in Sec. 3 that enforce the initial condition of the dynamics. The second one is due to the presence of the kick. As it emerges from Eqs. (96) and (97), while introducing the effect of the kick for one time quantity is straightforward, two point functions should incorporate at any $t' > 0$ the effect of the kick from $t = 0$. However the used numerical approach (see an example of source code at https://github.com/sphinxteam/spiked_matrix-tensor) obtains the two point correlation function $c(t,t')$ with $t > t'$ from, among other terms, the integration of $\{c(t-dt,t'')\}$ with $t'' \in [0, t-dt]$ (example in yellow in the Fig. 9). In this scheme only $c(t,t')$ with $t > t'$ are ever evaluated and used, but the singular contribution coming from the impulse at $t = 0$ would be only included in $c(dt,0)$, and all others contributions to $c(dt,t')$ would be lost. A solution to this issue has been implemented by evaluating separately $c(dt,t)$ with $dt < t$ through a modified integration routine on $\{c(0,t'')\}$ with $t'' \in [0,t]$ and

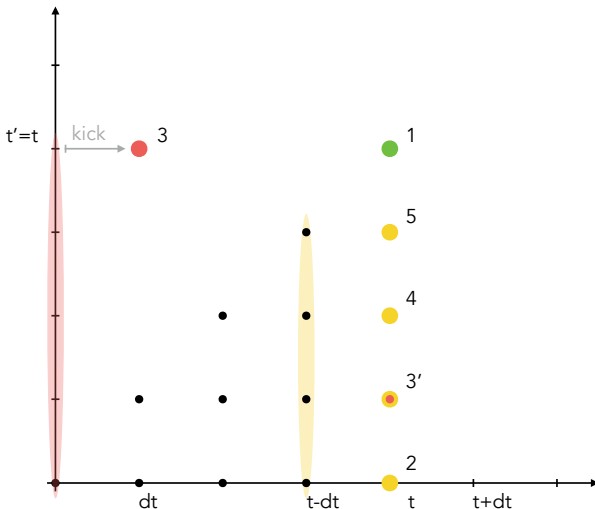

Figure 9: Integration scheme of two point functions proceeds imposing their values on the diagonal (green circle) to be 1 for correlation and 0 for response, and obtaining $c(t, t')$ and $r(t, t')$ (yellow circles) from the integration of the functions at $t - dt$ (shaded yellow area). Including the initial kick in the integration for $c(t, dt)$ (red dot) deserves a particular treatment. It is obtained by symmetry from $c(dt, t)$ (red circle) which is the result of integration of the functions $c(0, t')$ (shaded red area) plus the contribution from the kick.

adding the contribution from the kick. The result, by symmetry, gives $c(t, dt)$ (in red in Fig. 9) to be used in the subsequent integration step for $c(t + dt, t')$, which will then contain the contribution from the kick.

## 5.3 Free-fall dynamics from the saddle and asymptotic solution

We now present the full numerical solution with initial condition on the saddle $s_2$. The results shown in this section refer to a reference minimum $s_1$ at energy $\epsilon_1 = -1.167$ and initial condition on a saddle $s_2$ at overlap $q = 0.75$ from $s_1$ and at energy $\epsilon_2 = -1.1555$. We have taken $\alpha = 0$, *i.e.* zero temperature.

A first check of our numerical scheme is that without the kick the numerically integrated dynamics is stuck on the saddle, which is indeed what we find, as anticipated in Sec. 3.3.1. We then implement the kick as explained above and find the results reported in Fig. 10 in terms of the overlap $x(t)$ with the original minimum $s_1$ and the energy $\epsilon(t)$, for a positive and negative kick of amplitude $\varepsilon = 10^{-3}$. We observe that the dynamics on both sides of the saddle lead to a finite time relaxation towards the two neighboring minima. We validate the prediction for the long time energies and correlation obtained in Sec. 4 under the TTI (Time Translation Invariance) hypothesis, see the perfect correspondence in Fig. 10 with the long time limit of numerical integration for the corresponding quantities. We have also verified explicitly that TTI holds for correlation and response asymptotically (only the latter has a non-trivial TTI dynamics since $\alpha = 0$).

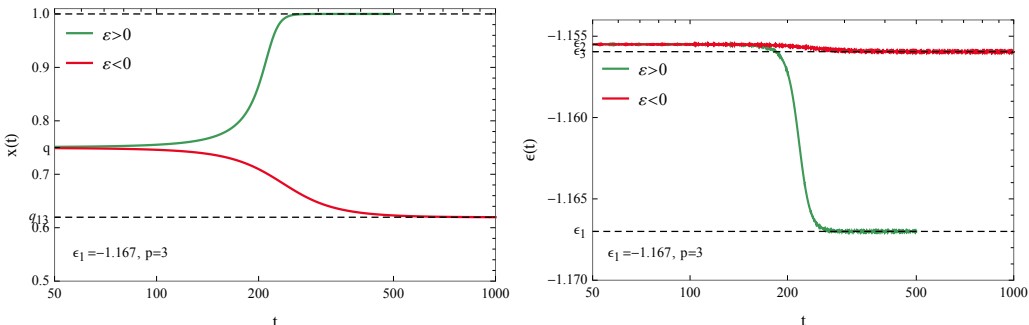

Figure 10: Results of the numerical integration of dynamical relaxation with kick of positive (negative) amplitude are represented in green (red). *Left.* Overlaps $x(t)$ between the reference minimum $s_1$ and the configuration along the dynamics. It is $x(t=0) = 0.75 = q$. At large time it approaches asymptotic values 1 and $q_{13} = 0.619584$ predicted in Sec. 4. *Right.* Energies along the relaxation paths start from the energy of the saddle $\epsilon_2 = -1.1555$ and reach $\epsilon_1 = -1.167$ and $\epsilon_3 = -1.15595$ predicted in Sec. 4.

## 6 The shape of the dynamical instanton

In this section we focus on the dynamical instanton, which corresponds to the activated process that allows the system to escape from the minimum $s_1$ to the new minimum $s_3$ by crossing the barrier associated to $s_2$. In order to obtain the dynamical instanton, we combine the results on free-fall dynamics derived above with time-reversal transformations. In fact, the theory of activated process at low temperature developed in theoretical physics and mathematics (referred to Freidlin and Wentzell in probability theory) established that an activated process can be decomposes in two parts: first an upward trajectory to the saddle, which is the time-reversal of the free-fall descent (in our case from $s_2$ to $s_1$), and then the free-fall descent from $s_2$ to the new minimum. In the following we recall the time-reversal field transformations that will allow us to reconstruct the dynamical instanton.

### 6.1 Time reversal

The time reversal $c_R(t,t')$, $r_R(t,t')$ of the correlation $c(t,t')$ and the response function $r(t,t')$ for $t > t'$ follows from the relation between the time reversal fields $s_R(t)$, $\hat{s}_R(t)$ and the original field $s(t)$ and auxiliary field $\hat{s}(t)$. Let us recall them [34, 46] in a simplified setting where

$$Z = \int \mathcal{D}s^t \mathcal{D}\hat{s}^t \, e^{S[s,\hat{s};\tau]}, \tag{98}$$

with an action

$$S[s,\hat{s};\tau] = \int_0^\tau dt \, \hat{s}(t) \left[ \frac{\alpha}{2} \hat{s}(t) - \frac{ds(t)}{dt} - \frac{\delta \mathcal{R}[s^t]}{\delta s(t)} \right], \tag{99}$$

and with $\mathcal{R}[s^t] = \mathcal{E}[s^t] + z(t)s(t)/2$. The single path time-reversal is as follows

$$s_R(t) = s(\tau - t) \tag{100}$$

$$\hat{s}_R(t) = \hat{s}(\tau - t) + \frac{2}{\alpha} \frac{ds(\tau - t)}{dt}.$$

This choice is self-explanatory for $s_R(t)$. The non trivial transformation of the auxiliary field is obtained instead by imposing the invariance under time inversion of the action in Eq. (99),

except from the production of boundary terms at $s(0) = s_I = s_R(\tau)$ and $s(\tau) = s_F = s_R(0)$ that assure detailed balance all along the dynamical path:

$$P[s(\tau)|s_I] = \int \mathcal{D}\hat{s}^t \, e^{S[s,\hat{s};\tau]} = P[s_R(\tau)|s_F]\exp\left[-\frac{2}{\alpha}(\mathcal{R}(s_F) - \mathcal{R}(s_I))\right]. \tag{101}$$

The transformations under time reversal for correlation and response functions, as defined in Eq. (25), are therefore inherited from the single field transformations as follows

$$c_R(t,t') = \lim_{N\to\infty} \frac{s_R(t) \cdot s_R(t')}{N} = \lim_{N\to\infty} \frac{s(\tau-t) \cdot s(\tau-t')}{N} = c(\tau-t, \tau-t') \tag{102}$$

$$\begin{aligned} r_R(t,t') &= \lim_{N\to\infty} \frac{s_R(t) \cdot \hat{s}_R(t')}{N} = \lim_{N\to\infty} \frac{s(\tau-t) \cdot (\hat{s}(\tau-t') + \frac{2}{\alpha}\frac{ds(\tau-t')}{dt'})}{N} \\ &= r(\tau-t, \tau-t') + \frac{2}{\alpha}\frac{d}{dt'}c(\tau-t, \tau-t') \end{aligned} \tag{103}$$

$$\begin{aligned} d_R(t,t') &= \lim_{N\to\infty} \frac{\hat{s}_R(t) \cdot \hat{s}_R(t')}{N} = \lim_{N\to\infty} \frac{(\hat{s}(\tau-t) + \frac{2}{\alpha}\frac{ds(\tau-t)}{dt}) \cdot (\hat{s}(\tau-t') + \frac{2}{\alpha}\frac{ds(\tau-t')}{dt'})}{N} \\ &= \frac{2}{\alpha}\left[\frac{d}{dt}r(\tau-t, \tau-t') + \frac{d}{dt'}r(\tau-t', \tau-t)\right] + \frac{4}{\alpha^2}\frac{d^2}{dt\,dt'}c(\tau-t, \tau-t'), \end{aligned} \tag{104}$$

as $d(t,t') = \lim_{N\to\infty} \hat{s}(t) \cdot \hat{s}(t')/N = 0$.

## 6.2 Reconstruction of the dynamical instanton

As schematically shown in Fig. 8, since we know by direct numerical integration the correlation and response function along the free-fall dynamics $s_2 \to s_1$, we can obtain their time-reversed counterparts using the relations above. We shall denote the corresponding correlation function $c_{\text{up}}(t,t')$ and the associated dynamical path $\mathcal{P}_{\text{up}}$. In order to construct the dynamical instanton, the time-reversed path thus obtained is merged with the forward dynamical path $\mathcal{P}_{\text{do}}$ from the saddle to the new minimum $s_3$. Accordingly, the correlation functions $c_{\text{do}}(t,t')$ for this process is obtained by direct numerical integration along the free-fall dynamics $s_2 \to s_3$. In the large $N$ limit, both these free-fall dynamics need infinite time, $\tau_{\text{up}}$ and $\tau_{\text{do}}$, to take place, but thanks to the introduction of the kick they can be visualised in a finite time window. Moreover, the probability rate of such dynamical instanton equals at leading order $e^{-2(E_2-E_1)/\alpha}$, with $E_2$ and $E_1$ the energy of the saddle and the original minimum, respectively, as it follows from the results recalled in the previous section. Since the difference in energy between the saddle and the original minimum is extensive, this implies that the activated process associated to the dynamical instanton typically takes place on a time-scale that diverges exponentially with $N$. We wish to describe the reconstructed dynamical instanton in terms of a global two time correlation function $c(t,t')$ defined on the entire time span $t \in [0, \tau_f]$ and $t' \in [0, \tau_f]$ where $\tau_f = \tau_{\text{up}} + \tau_{\text{do}}$, and $\tau_{\text{up}}$, $\tau_{\text{do}}$ are the time span of the dynamical paths respectively towards ($\mathcal{P}_{\text{up}}$) and from ($\mathcal{P}_{\text{do}}$) the saddle. Finally $\tau_s = \tau_{\text{up}}$ is the time at which the saddle is visited. However, a reconstruction based on a junction at time $\tau_s$ of these two distinct dynamical paths lacks the off-diagonal sectors where $t \in [\tau_s, \tau_f]$ and $t' \in [0, \tau_s]$, and viceversa. To fill this gap we propose an approximated interpolation of the correlation function $c(t,t')$ in these dynamical sectors based on the following decomposition for $t > \tau_s$ and $t' < \tau_s$

$$s(t) = s_2 \, c_{\text{do}}(t, \tau_s) + v\sqrt{1 - c_{\text{do}}^2(t, \tau_s)}, \tag{105}$$

$$s(t') = s_2 \, c_{\text{up}}(\tau_{\text{s}}, t') + v' \sqrt{1 - c_{\text{up}}^2(\tau_{\text{s}}, t')} \,, \tag{106}$$

with $v$ and $v'$ two vectors on the sphere, perpendicular to $s_2$. For $t$ and $t'$ approaching $\tau_{\text{s}}$ both vectors correspond to the saddle $s_2$. The above decomposition corresponds to fixing the projection of the dynamical variables $s(t)$, $s(t')$ along the direction of the saddle to its typical value, which is given by the solution of the dynamical equations. The projection along the orthogonal direction is then automatically fixed by the spherical constraint. The directions $v$ and $v'$ are in principle varying with time during the dynamical evolution, and so is their overlap. We neglect this time dependence and set:

$$\lim_{N \to \infty} \frac{v \cdot v'}{N} = \frac{q_{13} - q \, q_{23}}{\sqrt{1 - q^2} \sqrt{1 - q_{23}^2}} \,. \tag{107}$$

This condition ensures that the boundary conditions are verified: at $t = \tau_{\text{f}}$, $t' = 0$, where it is expected $s(\tau_{\text{f}}) = s_3$, $s(0) = s_1$, we have that their scalar product is $q_{13}$ as it should, since $c_{\text{do}}(\tau_{\text{f}}, \tau_{\text{s}}) = q_{23}$, $c_{\text{up}}(\tau_{\text{s}}, 0) = q$.

The resulting expression for the correlation function for $t \in [\tau_{\text{s}}, \tau_{\text{f}}]$ and $t' \in [0, \tau_{\text{s}}]$ then reads

$$c(t, t') = c_{\text{do}}(t, \tau_{\text{s}}) c_{\text{up}}(\tau_{\text{s}}, t') + \frac{q_{13} - q q_{23}}{\sqrt{1 - q^2} \sqrt{1 - q_{23}^2}} \sqrt{1 - c_{\text{do}}(t, \tau_{\text{s}})^2} \sqrt{1 - c_{\text{up}}(\tau_{\text{s}}, t')^2} \,. \tag{108}$$

Finally we get $c(t', t) = c(t, t')$ by symmetry. We are now in position to completely reconstruct the dynamical instanton corresponding to barrier crossing in mean-field glassy systems. Its shape is shown in Fig. 4.

# 7 Conclusion

The main outcome of this work is the identification, for a prototypical fully-connected model of glasses, of the simplest activated processes, which correspond to the escape from a given minimum through the family of saddles of index one that are closer to the minimum in configuration space. By combining the Kac-Rice method and dynamical field theory, we have constructed explicitly the dynamical instanton associated to the jump over the barrier, and characterized the new minima that the system can reach after the jump. We have found that the minima that are reached dynamically through these saddles are at extensively higher energy than the reference one and are strongly correlated to it, being relatively close in configurations space. This allows us to get some insight on the kind of dynamics one should expect in the activated regime, at least for the particular model we are considering: indeed, it is natural to expect that these saddles will matter in the earlier times of the dynamics, and that escaping through them the system would undergo a back and forth motion with frequent returns to the original minimum, given that the energy barrier associated to going back to the reference minimum is extensively lower. Such frequent returns have been recently observed in numerical simulations of the low-temperature dynamics of the Ising $p$-spin model of finite-size [13, 47]. In [47] in particular it is shown that most of the stable configurations (the analogous of local minima in a discrete setting) that the system visits consecutively in its activated dynamics have a large overlap with each others; moreover, it appears that the system has to climb higher in the energy landscape in order to reach stable configurations that are less correlated with the previous one, consistently with our finding that the minima at smaller overlap with the reference one are connected to it by saddles at higher energy density (at least when focusing on the saddles at larger overlap and zero complexity, see Fig. 6).

Let us comment on the role of the dimensionality $N$ and of the temperature $T$ in our analysis. As we have stressed in Sec. 2.3, our calculation differs with respect to a standard instanton calculation performed minimizing a suitably-defined large deviation (dynamical) functional. We use the knowledge gained from the Kac-Rice analysis about the index-1 saddles around one given minimum and study the relaxation dynamics starting from a given saddle. The relaxation from the saddle to the minima is a *typical* dynamical process, and thus it does not require to compute large deviations of the dynamical functionals. The instanton is then obtained by time-reversing the solution which goes back to the original minimum. This approach allows us to obtain insights on the dynamics at times scaling exponentially with the system size $N$ and does not require to solve the very challenging problem of finding non-casual solutions associated to the large deviation (dynamical) functional [36]. Note that in our analysis we do not take into account the finite $N$ corrections to the landscape statistics; analyzing how those affect dynamics at finite times remains a challenging open question.

For what concerns the role of temperature, the initial conditions of our dynamical equations are unstable stationary points of the *energy* landscape; moreover, we solved the dynamical equations setting $\alpha = 0$, thus describing gradient descent from the saddle to a nearby minimum of the energy landscape. The instantons we obtain are therefore expected to capture the dynamics at very small values of temperature: small enough so that the energy landscape is a good approximation of the free-energy one, but non-zero, so that barrier-crossing is a possible even though extremely rare process. This is particularly meaningful for the spherical $p$-spin model, given that the free-energy landscape has a continuous dependence on temperature in that model. For more generic models, the relevant landscape at finite temperatures is the free-energy one. In the fully-connected limit, the free-energy landscape of the system can be characterized in terms of the so called TAP functional $f_{\text{TAP}}(\mathbf{m})$ [48], depending on the local magnetizations $\mathbf{m} = (m_1, \cdots, m_N)$ rather than on the spin configurations $\mathbf{s}$. The stationary points of this functional have a physical meaning: stable local minima can be identified with the system's metastable states; unstable saddles are also attractors of the dynamics, when thought of as an evolution on the free energy surface [49]. Therefore, one can envisage a dynamical calculation similar to the one presented in this work, with the TAP free energy replacing the energy landscape. The special instantonic solution found in [19] shows in a concrete example that using the TAP landscape to analyze thermal activation is justified. Note that this finite temperature treatment could be particularly interesting to pursue for models in which thermal fluctuations modify substantially the landscape giving rise to a chaotic dependence on temperature [50–52] (see Ref. [53] for recent progress in the characterization of these landscapes).

To conclude, the results presented in this work represent a first step towards a general classification and analysis of dynamical instantons in rough high-dimensional energy landscapes. In particular, the dynamical equations we derived allow us to describe escapes from local minima passing through a particular family of index-1 saddles, those that are closer to the reference minimum in configuration space. The reason for this is that in this range of overlaps, the index-1 saddles are the typical stationary points (i.e., those that are exponentially more numerous than any other type of stationary points). Other saddles geometrically connected to the reference minimum exist at higher distance (smaller value of the overlap) but are atypical [23], i.e. they are still exponentially numerous in $N$, but their number is subleading with respect to that of local minima, which are instead the typical critical points for that value of the overlap. Initializing the dynamics in one of these saddles requires to condition explicitly on the properties of the Hessian of the initial condition, thus generating additional terms in the dynamical equations. Deriving the corresponding dynamical equations and characterizing their asymptotic solutions is potentially interesting, since these saddles might connect the reference minimum to other local minima that are less correlated with the reference one,

being at smaller overlap with it or having lower energy. These saddles might provide more direct escape paths, less affected by the frequent returns mentioned above. We leave this interesting open problem to future work. More broadly, it is worth examining the extremization equation of the large deviation dynamical functional by leveraging on the special solution we constructed. Generalizing such solution (numerically or analytically) provides a new way to obtain the dynamical instantons which correspond to more complex activated processes, and in particular the ones leading to thermal relaxation.

# Acknowledgements

We acknowledge Stefano Sarao Mannelli and Pierfrancesco Urbani for sharing the original version of the code, available at https://github.com/sphinxteam/spiked_matrix-tensor. We also thank J. Kurchan and G. Tarjus for interesting discussions.

**Funding information** This work is supported by the Simons Foundation collaboration Cracking the Glass Problem (No. 454935 to G. Biroli). V.Ros acknowledges funding by the LabEx ENS-ICFP: ANR-10-LABX-0010/ANR-10-IDEX-0001-02 PSL*

# A   Statistics of the Hessian at critical points

In this Appendix we recall the statistics of the Hessian matrices of the functional (1), evaluated at stationary points $s_2$ that are at fixed overlap $q$ from a reference minimum $s_1$. This statistics has been computed in [22] (see also Lemma 13 in [33] and [23]), and we refer to that work for the details of the derivation. For a fixed realization of the random field, the Hessian matrix $\mathcal{H}[s]$ at an arbitrary point $s$ on the sphere is given by (12): the first contribution is simply the projection of the matrix of second derivatives of $\mathcal{E}[s]$ into the tangent plane at $s$, while the second term comes from enforcing the spherical constraint. Conditioning on $\mathcal{E}[s] = N\epsilon$, we see that the Hessian can be re-written as:

$$\mathcal{H}_{\alpha\beta} = \left( e_\alpha[s] \cdot \frac{\delta^2 \mathcal{E}[s]}{\delta s^2} \cdot e_\beta[s] - p\,\epsilon\,\delta_{\alpha\beta} \right), \tag{109}$$

where the vectors $e_\alpha[s]$ form a basis of the tangent plane at $s$. Following the notation in [22,23] we focus on the rescaled matrix:

$$\mathcal{M}_{\alpha\beta}[s_2] = \sqrt{2N} \left( e_\alpha[s_2] \cdot \frac{\delta^2 \mathcal{E}[s_2]}{\delta s^2} \cdot e_\beta[s_2] \right), \tag{110}$$

and describe the statistics of its entries averaged over the random couplings $J_{i_1,\cdots,i_p}$, once conditioning the point $s_2$ to be a stationary point at overlap $q$ from another stationary point $s_1$ with energy density $\epsilon_1$. To do so, we choose the basis vectors $e_\alpha[s_2]$ in such a way that only the vector $e_{N-1}[s_2]$ has a non-zero projection on $s_1$,

$$e_{N-1}[s_2] = \frac{s_1 - q s_2}{\sqrt{N[1-q^2]}}, \tag{111}$$

while the remaining $e_\alpha$ for $\alpha \leq N-2$ span the region of space that is orthogonal to both $s_1$ and $s_2$. The statistics of the conditioned matrix $\mathcal{M}$ is invariant with respect to the particular choice of these $N-2$ vectors: the entries $\mathcal{M}_{\alpha\beta}$ with $\alpha, \beta \leq N-2$ are independent Gaussian variables with variance $\sigma^2 = p(p-1)$, and thus form a huge block with GOE statistics. The

entries $\mathcal{M}_{\alpha N-1}$ with $\alpha \neq N-1$ are again independent Gaussian variables, but have a modified variance:

$$\Delta^2(q) \equiv \sigma^2 \left( 1 - \frac{(p-1)(1-q^2)q^{2p-4}}{1-q^{2p-2}} \right). \tag{112}$$

Finally, the diagonal element $\mathcal{M}_{N-1N-1}$ has yet another variance (that we do not report since it is not relevant in the following), and a non-zero average equal to:

$$\langle \mathcal{M}_{N-1N-1} \rangle = \sqrt{N}\mu(q,\epsilon_1,\epsilon_2) \equiv \frac{\sqrt{2N}[\epsilon_1 a_0(q) - \epsilon_2 a_1(q)]}{a_2(2)}, \tag{113}$$

with the constants $a_i(q)$ already defined in (38) in the main text. Therefore, $\mathcal{M}$ is a GOE matrix modified by finite-rank additive and multiplicative perturbations that alter the statistics of the entries in the last line and column, that single out the direction connecting $s_1$ and $s_2$ in configuration space. The bulk of the eigenvalue density of $\mathcal{M}$ is given by a semicircle, and it is insensitive to the modified statistics of the elements outside the $(N-2) \times (N-2)$ invariant block. As argued in [22,23], the perturbations to the GOE statistics can nevertheless generate a sub-leading correction to this density, in the form of a single isolated eigenvalue $\lambda_{\min}(q,\epsilon,\epsilon_0)$ that lies outside the support of the semicircle. This eigenvalue exists whenever [23]

$$\mu < -\sigma\left[1 + \frac{(\sigma')^2}{\sigma^2}\right] \qquad \text{where} \qquad \sigma'(q) = \sqrt{\sigma^2 - \Delta^2(q)}, \tag{114}$$

and it solves the equation

$$\lambda - \mu(q,\epsilon,\epsilon_0) - \Delta^2(q)G_\sigma(\lambda) = 0, \tag{115}$$

with [7]

$$G_\sigma(x) = \frac{1}{2\sigma^2}\left(x - \text{sign}(x)\sqrt{x^2 - 4\sigma^2}\right). \tag{116}$$

The solution to this equation can be compactly written as:

$$\lambda_{\min}(q,\epsilon_1,\epsilon_2) = G_\sigma^{-1}\left(G_{\sigma'}(\mu)\right) = \frac{1}{G_{\sigma'}(\mu)} + \sigma^2 G_{\sigma'}(\mu) \qquad \text{with} \qquad G_\sigma^{-1}(x) = \frac{1}{x} + \sigma^2 x. \tag{117}$$

When (114) holds and when the smallest eigenvalue of the matrix $\sqrt{2}\mathcal{H}$,

$$\lambda_0(q,\epsilon_1,\epsilon_2) \equiv \lambda_{\min}(q,\epsilon_1,\epsilon_2) - \sqrt{2}p\epsilon_2, \tag{118}$$

is negative, the point $s_2$ is an index-1 saddle. The eigenvector associated to this eigenvalue has a macroscopic projection along the direction in configuration space connecting the saddle $s_2$ to the minimum $s_1$ (see [23] for the explicit calculation of the magnitude of this projection). This is what happens for parameters that correspond to the violet region in figure 2.

## B   Derivation of Eq. 23

In this Appendix, we derive Eq. (23). We introduce the shorthand notation $M[s_a^t] \equiv M^a(t)$ and enforce the initial conditions as:

$$\int \prod_{i \leq j=1}^{N} dm_{ij}^1 dm_{ij}^2 \int \prod_{i \leq j=1}^{N} d\lambda_{ij}^1 d\lambda_{ij}^2 e^{i\lambda_{ij}^1\left(M_{ij}^1(0)-m_{ij}^1\right)} e^{i\lambda_{ij}^2\left(M_{ij}^2(0)-m_{ij}^2\right)}. \tag{119}$$

---

[7]The sign in front of the square root of $G_\sigma(x)$ guarantees that the resolvent is positive for $x > 0$, and decays to zero as $|x| \to \infty$.

We can therefore re-write the average (17) as

$$
\mathcal{J} = \int \prod_{a=1}^{2} \prod_{i \leq j=1}^{N} \left[ dm_{ij}^a \, d\lambda_{ij}^a e^{-i\lambda_{ij}^a m_{ij}^a} \right] \mathcal{F}[m^a, s_a] \int \mathcal{D}M^a e^{-\sum\limits_{a=1}^{2} \sum\limits_{i \leq j}^{N} \int_0^\infty dt \, M_{ij}^a(t) \left[ \delta_{a,2} O_{ij}(t) - 2i\lambda_{ij}^a \delta(t) \right]},
$$

(120)

where $\mathcal{D}M^a$ denotes the joint Gaussian measure:

$$
\mathcal{D}M^a = \mathcal{D}[M_{ij}^a(t)] \exp \left\{ -\frac{1}{2} \sum_{a=1}^{2} \sum_{i \leq j} \sum_{k \leq l} \int_0^\infty dt \int_0^\infty dt' \, M_{ij}^a(t) [\Sigma^{-1}]_{ij,kl}^{ab}(t, t') M_{kl}^b(t') \right\},
$$

(121)

and given that $M_{ij}(t)$ is symmetric we have restricted the covariance matrix to $i \leq j$ and $k \leq l$:

$$
\Sigma_{ij,kl}^{ab}(t, t') \equiv \chi_{i \leq j} \, \chi_{k \leq l} \, \langle M_{ij}^a(t) M_{kl}^b(t') \rangle,
$$

(122)

where $\chi$ is an indicator function. The matrix at the exponent in (120) reads

$$
O_{ij}(t) = \frac{1}{p-1} \left\{ [\hat{s}_2(t)]_i [s_2(t)]_j + [\hat{s}_2(t)]_j [s_2(t)]_i - \delta_{ij} [\hat{s}_2(t)]_i [s_2(t)]_i \right\},
$$

(123)

and:

$$
\mathcal{F}[m^a, s_a] = \prod_{a=1}^{2} \prod_{\alpha=1}^{N-1} \delta \left( \frac{e^\alpha[s_a] \cdot m^a \cdot s_a}{p-1} \right) \delta \left( \frac{s_a \cdot m^a \cdot s_a}{p(p-1)} - N\epsilon_a \right) \left| \det \left( \overline{m}^a - p \frac{\mathcal{E}[s_a]}{N} \mathbb{1} \right) \right|,
$$

(124)

where $\overline{m}^a$ is the projection of the matrix $m^a$ onto the tangent plane at $s_a$, and $\mathbb{1}$ is the identity matrix. Notice that the fields in (123) are exactly at equal time: this will be relevant for the discussion in Appendix C. The Gaussian integration over the matrix field and over the auxiliary variables $\lambda_{ij}^a$ (both Gaussian) gives for (16) the following expression:

$$
\mathcal{I}(\epsilon_2, q | \epsilon_1) \propto \int_{s_1 \cdot s_2 = Nq} ds_1 ds_2 \int \prod_{a=1}^{2} \prod_{i \leq j=1}^{N} dm_{ij}^a \mathcal{F}[m^a, s_a] \int_{s_2(0)=s_2} \mathcal{D}s^t \mathcal{D}\hat{s}^t \, e^{\mathcal{V}_0 + \mathcal{V}},
$$

(125)

with an action

$$
\mathcal{V} = \frac{1}{2} \sum_{i \leq j} \sum_{k \leq l} \left\{ \int_0^\infty dt \int_0^\infty dt' O_{ij}(t) \Sigma_{ij,kl}^{22}(t, t') O_{kl}(t') - \left( \Xi_{ij}^a + m_{ij}^a \right) \Omega_{ij,kl}^{ab} (\Xi_{kl}^b + m_{kl}^b) \right\},
$$

(126)

where

$$
\Omega = [\Sigma(0,0)]^{-1}, \qquad \Xi_{ij}^a = \int_0^\infty dt \, \Sigma_{ij,kl}^{a2}(0, t) O_{kl}(t).
$$

(127)

The proportionality is due to the fact that we are neglecting the functional determinant arising from the integration over the matrix field, as well as the determinant resulting from the Gaussian integration over $\lambda_{ij}$. These terms can be disregarded as they do not depend explicitly on the spin variables, and therefore will not matter when deriving the dynamical equations from the optimization of the dynamical action. The expression for $\mathcal{V}_0$ is given in (18).

We now focus on the integration over the initial conditions $m_{ij}^a$. In order to implement the constraints in (124), it is convenient to express the components of the matrices $m^a$ in the bases $e^\alpha[s_a]$ in which the constraints are given, which span the tangent planes to the sphere at

$s_a$. To this aim, we introduce the rescaled unit vectors $\sigma_a = s_a/\sqrt{N}$ for $a = 1, 2$. We introduce a first set of unit vectors $\mathcal{B}_1 = \{e_1, \cdots, e_{N-2}, w_{N-1}, w_N\}$ such that:

$$w_N = \sigma_1, \qquad w_{N-1} = \frac{\sigma_2 - q\sigma_1}{\sqrt{1-q^2}}, \tag{128}$$

and the remaining $e_\alpha$ for $\alpha \leq N-2$ span the region of space that is orthogonal to both $s_1$ and $s_2$. Analogously, we introduce a second set $\mathcal{B}_2 = \{e_1, \cdots, e_{N-2}, v_{N-1}, v_N\}$ such that:

$$v_N = \sigma_2, \qquad v_{N-1} = \frac{\sigma_1 - q\sigma_2}{\sqrt{1-q^2}}. \tag{129}$$

These two sets are related by:

$$\begin{pmatrix} v_{N-1} \\ v_N \end{pmatrix} = \begin{pmatrix} -q & \sqrt{1-q^2} \\ \sqrt{1-q^2} & q \end{pmatrix} \begin{pmatrix} w_{N-1} \\ w_N \end{pmatrix}. \tag{130}$$

The vectors $e^\alpha[s_1]$ in (124) spanning the tangent plane at $s_1$ can be chosen to be equal to $\mathcal{B}_1 \backslash \{w_N\}$, while the vectors $e^\alpha[s_2]$ can be identified with $\mathcal{B}_2 \backslash \{v_N\}$. It is convenient to determine the covariances (122) between the matrix elements $M^a_{\alpha\beta}$ expressed in the corresponding bases $\mathcal{B}_a$. For $K = N-2$, let us collect the matrix elements $M^a_{\alpha\beta}$ into the following vectors:

$$\begin{aligned}
\vec{M}_0 &= (M^1_{11}, M^2_{11}, \cdots, M^1_{KK}, M^2_{KK}, M^1_{12}, M^2_{12}, \cdots, M^1_{K-1K}, M^2_{K-1K}) \\
\vec{M}_{1/2} &= (M^1_{1N-1}, M^1_{1N}, M^2_{1N-1}, M^2_{1N}, \cdots, M^1_{KN-1}, M^1_{KN}, M^2_{KN-1}, M^2_{KN}) \\
\vec{M}_1 &= (M^1_{N-1N-1}, M^1_{NN}, M^1_{N-1N}, M^2_{N-1N-1}, M^2_{NN}, M^2_{N-1N}).
\end{aligned} \tag{131}$$

It is easy to check that at $t = 0 = t'$ the covariance matrix $\Sigma \equiv \Sigma(0,0)$, and thus its inverse $\Omega$, have a block-diagonal structure with respect to this decomposition:

$$\Sigma(0,0) = \begin{pmatrix} \Sigma_0 & 0 & 0 \\ 0 & \Sigma_{1/2} & 0 \\ 0 & 0 & \Sigma_1 \end{pmatrix} \longrightarrow \Omega = [\Sigma(0,0)]^{-1} = \begin{pmatrix} \Omega_0 & 0 & 0 \\ 0 & \Omega_{1/2} & 0 \\ 0 & 0 & \Omega_1 \end{pmatrix}. \tag{132}$$

Let us determine the explicit form of $\Sigma(0,0)$. The first block has a particularly simple structure:

$$[\Sigma_0]^{ab}_{\alpha\beta,\gamma\delta} = \langle M^a_{\alpha\beta} M^b_{\gamma\delta} \rangle = \delta_{\alpha\gamma}\delta_{\beta\delta} \frac{p(p-1)}{2N}(\sigma_a \cdot \sigma_b)^{p-2}(1 + \delta_{\alpha\beta}) \quad \text{for} \quad \alpha, \beta, \gamma, \delta \leq K = N-2, \tag{133}$$

indicating that the first $(N-2) \times (N-2)$-dimensional blocks of the matrices $M^a$ have a coupled GOE statistics: each $M^a_{\alpha\beta}$ is correlated only with itself and with the corresponding entry $M^b_{\alpha\beta}$ of the other matrix. For what concerns the correlations between the components in $\vec{M}_{1/2}$, it can be easily shown that $\langle M^a_{\alpha x} M^b_{\gamma y} \rangle \propto \delta_{\alpha\gamma}$ for $\alpha, \gamma \leq N-2$ and $x, y \in \{N-1, N\}$. The blocks in the covariance matrix have the same form for each $\alpha$:

$$\begin{pmatrix} \Sigma_{N-1N-1} & \Sigma_{N-1N} \\ \Sigma_{NN-1} & \Sigma_{NN} \end{pmatrix} \equiv \begin{pmatrix} \langle M^1_{\alpha N-1} M^1_{\alpha N-1} \rangle & \langle M^1_{\alpha N-1} M^2_{\alpha N-1} \rangle & \langle M^1_{\alpha N-1} M^1_{\alpha N} \rangle & \langle M^1_{\alpha N-1} M^2_{\alpha N} \rangle \\ \langle M^2_{\alpha N-1} M^1_{\alpha N-1} \rangle & \langle M^2_{\alpha N-1} M^2_{\alpha N-1} \rangle & \langle M^2_{\alpha N-1} M^1_{\alpha N} \rangle & \langle M^2_{\alpha N-1} M^2_{\alpha N} \rangle \\ \langle M^1_{\alpha N} M^1_{\alpha N-1} \rangle & \langle M^1_{\alpha N} M^2_{\alpha N-1} \rangle & \langle M^1_{\alpha N} M^1_{\alpha N} \rangle & \langle M^1_{\alpha N} M^2_{\alpha N} \rangle \\ \langle M^2_{\alpha N} M^1_{\alpha N-1} \rangle & \langle M^2_{\alpha N} M^2_{\alpha N-1} \rangle & \langle M^2_{\alpha N} M^1_{\alpha N} \rangle & \langle M^2_{\alpha N} M^2_{\alpha N} \rangle \end{pmatrix}$$

$$= \frac{p(p-1)}{2N} \times$$

$$\times \begin{pmatrix} 1 & q^{p-3}(-pq^2+q^2+p-2) & 0 & (p-1)q^{p-2}\sqrt{1-q^2} \\ q^{p-3}(-pq^2+q^2+p-2) & 1 & (p-1)q^{p-2}\sqrt{1-q^2} & 0 \\ 0 & (p-1)q^{p-2}\sqrt{1-q^2} & p-1 & (p-1)q^{p-1} \\ (p-1)q^{p-2}\sqrt{1-q^2} & 0 & (p-1)q^{p-1} & p-1 \end{pmatrix}, \tag{134}$$

where we introduced the compact notation $\Sigma_{NN}$ for the $2 \times 2$ matrices with components $\Sigma^{ab}_{\alpha N, \alpha N}$, which are equal for any $\alpha \leq N - 2$, and similarly for the other blocks. Notice that this reduces to a diagonal matrix for $q \to 0$, when the initial condition $s_2$ of the dynamics is orthogonal (and thus uncorrelated) to the minimum $s_1$ [8]. Finally, the correlations of the components of $\vec{M}_1$ form a $6 \times 6$ matrix with the following block structure:

$$\Sigma_1 = \frac{p(p-1)}{2N} \begin{pmatrix} \Sigma_{N-1N-1,N-1N-1} & \Sigma_{N-1N-1,N-1N} & \Sigma_{N-1N-1,NN} \\ \Sigma_{N-1N,N-1N-1} & \Sigma_{N-1N,N-1N} & \Sigma_{N-1N,NN} \\ \Sigma_{NN,N-1N-1} & \Sigma_{NN,N-1N} & \Sigma_{NN,NN} \end{pmatrix}, \tag{135}$$

where each block is a $2 \times 2$ matrix with components $\Sigma^{ab}_{xy,z\xi} = \langle M^a_{xy} M^b_{z\xi} \rangle$ and $x, y, z, \xi \in \{N-1, N\}$. The various block read:

$$\Sigma_{N-1N-1,N-1N-1} = \begin{pmatrix} 2 & a \\ a & 2 \end{pmatrix}, \quad \Sigma_{N-1N-1,NN} = \begin{pmatrix} 0 & b \\ b & 0 \end{pmatrix}, \quad \Sigma_{N-1N-1,N-1N} = \begin{pmatrix} 0 & c \\ c & 0 \end{pmatrix}$$

$$\Sigma_{NN,NN} = p(p-1) \begin{pmatrix} 1 & q^p \\ q^p & 1 \end{pmatrix}, \quad \Sigma_{NN,N-1N} = \begin{pmatrix} 0 & d \\ d & 0 \end{pmatrix}, \quad \Sigma_{N-1N,N-1N} = (p-1) \begin{pmatrix} 1 & f \\ f & 1 \end{pmatrix}, \tag{136}$$

with

$$\begin{aligned} a &= q^{p-4} \left( (p-1)pq^4 - 2(p-2)(p-1)q^2 + p^2 - 5p + 6 \right) \\ b &= (p-1)pq^{p-2} \left( 1 - q^2 \right) \\ c &= -(p-1)q^{p-3} \sqrt{1-q^2} \left( p \left( q^2 - 1 \right) + 2 \right) \\ d &= p(p-1)q^{p-1} \sqrt{1-q^2} \\ f &= -q^{p-2} [1 - p(1-q^2)]. \end{aligned} \tag{137}$$

This general structure allows to decompose the sum in (126) in the following way:

$$\sum_{\alpha \leq \beta} \sum_{\gamma \leq \delta} \sum_{a,b=1}^{2} \left( \Xi^a_{\alpha\beta} + m^a_{\alpha\beta} \right) \Omega^{ab}_{\alpha\beta,\gamma\delta} (\Xi^b_{\gamma\delta} + m^b_{\gamma\delta}) = U_0 + U_{1/2} + U_1, \tag{138}$$

where:

$$\begin{aligned} U_0 &= \sum_{\alpha \leq \beta=1}^{N-2} \sum_{\gamma \leq \delta=1}^{N-2} \sum_{a,b=1}^{2} \left( \Xi^a_{\alpha\beta} + m^a_{\alpha\beta} \right) [\Omega_0]^{ab}_{\alpha\beta,\gamma\delta} (\Xi^b_{\gamma\delta} + m^b_{\gamma\delta}) \\ U_{1/2} &= \sum_{\alpha=1}^{N-2} \sum_{x,y=N-1}^{N} \sum_{a,b=1}^{2} \left( \Xi^a_{\alpha x} + m^a_{\alpha x} \right) [\Omega_{\frac{1}{2}}]^{ab}_{\alpha x,\alpha y} (\Xi^b_{\alpha y} + m^b_{\alpha y}) \\ U_1 &= \sum_{x,y,z,\xi=N-1}^{N} \sum_{a,b=1}^{2} \left( \Xi^a_{xy} + m^a_{xy} \right) [\Omega_1]^{ab}_{xy,z\xi} (\Xi^b_{z\xi} + m^b_{z\xi}). \end{aligned} \tag{139}$$

The constraints in (124) correspond to setting $m^a_{\alpha N} = 0$ for $\alpha < N$, and $m^a_{NN} = p(p-1)\epsilon_a$. Notice that the term $U_{1/2}$ couples the matrix elements $m^a_{\alpha N}$, that have to be set to zero, with the elements $m^a_{\alpha N-1}$, on which the integration is free. Similarly, the integration on the elements $m^a_{N-1N-1}$ in $U_1$ is free, while the elements $m^a_{N-1N}$ and $m^a_{NN}$ are constrained to take a given value. To decouple the constrained matrix elements from the unconstrained ones, we make use

---

[8] The case $p = 3$ has to be treated with more care, as in this case the off-diagonal matrix elements should be set to zero from the onset.

of Gaussian conditioning [9]. Introducing the vector notation $\Xi_{\alpha\beta} = (\Xi^1_{\alpha\beta}, \Xi^2_{\alpha\beta})^T$ and imposing $m^a_{\alpha N} = 0$, we obtain:

$$U_{1/2} \longrightarrow \sum_{\alpha=1}^{N-2} \Xi^T_{\alpha N} [\Sigma_{NN}]^{-1} \Xi_{\alpha N} + (m_{\alpha N-1} + \Xi^*_{\alpha 1})^T [\Sigma^*_{1/2}]^{-1} (m_{\alpha N-1} + \Xi^*_{\alpha 1}). \tag{142}$$

The second term in the sum (142) depend on some shifted 2-dimensional vectors $\Xi^*_{\alpha 1}$ and on a modified $2 \times 2$ correlation matrix $\Sigma^*_{1/2}$ given by:

$$\begin{aligned}
\Xi^*_{\alpha 1} &= \Xi_{\alpha N-1} - \Sigma_{N-1 N} \Sigma_{NN}^{-1} \Xi_{\alpha N}, \\
\Sigma^*_{1/2} &= \Sigma_{N-1 N-1} - \Sigma_{N-1 N} \Sigma_{NN}^{-1} \Sigma_{N N-1}.
\end{aligned} \tag{143}$$

We find:

$$\Sigma^*_{1/2} = \frac{p(p-1)}{2N} \begin{pmatrix} 1 - \frac{(p-1)(1-q^2)q^{2p-4}}{1-q^{2p-2}} & -\frac{q^{2p}-(p-1)q^4+(p-2)q^2}{q^{p+3}-q^{5-p}} \\ -\frac{q^{2p}-(p-1)q^4+(p-2)q^2}{q^{p+3}-q^{5-p}} & 1 - \frac{(p-1)(1-q^2)q^{2p-4}}{1-q^{2p-2}} \end{pmatrix}. \tag{144}$$

With an analogous reasoning, setting $\epsilon = (\epsilon_1, \epsilon_2)^T$, we see that once the conditioning on $m^a_{N-1 N} = 0$ and $m^a_{NN} = p(p-1)\epsilon_a$ are implemented the sum $U_1$ takes the form:

$$\begin{aligned}
U_1 \to &\begin{pmatrix} \Xi_{N-1 N} \\ \Xi_{NN} + p(p-1)\epsilon \end{pmatrix}^T [\Sigma_{\{1,0\}}]^{-1} \begin{pmatrix} \Xi_{N-1 N} \\ \Xi_{NN} + p(p-1)\epsilon \end{pmatrix} \\
&+ (m_{N-1 N-1} + \Xi^{**}_{11})^T [\Sigma^*_1]^{-1} (m_{N-1 N-1} + \Xi^{**}_{11}).
\end{aligned} \tag{145}$$

In this case $\Sigma_{\{1,0\}}$ is a shorthand notation for the $4 \times 4$ matrix with block structure:

$$\Sigma_{\{1,0\}} = \begin{pmatrix} \Sigma_{N-1 N, N-1 N} & \Sigma_{N-1 N, NN} \\ \Sigma_{NN, N-1 N} & \Sigma_{NN, NN} \end{pmatrix} = \frac{p(p-1)^2}{2N} \times$$

$$\begin{pmatrix} 1 & q^{p-2}[p(1-q^2)-1] & 0 & pq^{p-1}\sqrt{1-q^2} \\ q^{p-2}[p(1-q^2)-1] & 1 & pq^{p-1}\sqrt{1-q^2} & 0 \\ 0 & pq^{p-1}\sqrt{1-q^2} & p & pq^p \\ pq^{p-1}\sqrt{1-q^2} & 0 & pq^p & p \end{pmatrix}, \tag{146}$$

and:

$$\begin{aligned}
\Xi^{**}_{11} &= \Xi_{N-1 N-1} - (\Sigma_{N-1 N-1, N-1 N} \ \Sigma_{N-1 N-1, NN}) [\Sigma_{\{1,0\}}]^{-1} \begin{pmatrix} \Xi_{N-1 N} \\ \Xi_{NN} + p(p-1)\epsilon \end{pmatrix}, \\
\Sigma^*_1 &= \Sigma_{N-1 N-1, N-1 N-1} - (\Sigma_{N-1 N-1, N-1 N} \ \Sigma_{N-1 N-1, NN}) [\Sigma_{\{1,0\}}]^{-1} \begin{pmatrix} \Sigma_{N-1 N, N-1 N-1} \\ \Sigma_{NN, N-1 N-1} \end{pmatrix}.
\end{aligned} \tag{147}$$

Defining:

$$\mathcal{S}_0 = \frac{1}{2} \sum_{i \le j} \sum_{k \le l} \int_0^\infty dt \int_0^\infty dt' O_{ij}(t) \Sigma^{22}_{ij,kl}(t,t') O_{kl}(t'), \tag{148}$$

---

[9] We make use of the following identity holding for two generic vectors $x_1, x_2$:

$$\sum_{ij=1}^2 (x_i - \overline{x}_i)^T [\Sigma^{-1}]_{ij} (x_j - \overline{x}_j) = (x_2 - \overline{x}_2)^T [\Sigma_{22}]^{-1} (x_2 - \overline{x}_2) + (x_1 - \overline{x}^*_1(x_2))^T [\Sigma^*_{11}]^{-1} (x_1 - \overline{x}^*_1(x_2)), \tag{140}$$

where $\Sigma$ is a generic correlation matrix with blocks $\Sigma_{ij}$ and:

$$\Sigma^*_{11} = \Sigma_{11} - \Sigma_{12} \Sigma_{22}^{-1} \Sigma_{21}, \qquad \overline{x}^*_1(x_2) = \overline{x}_1 + \Sigma_{12} \Sigma_{22}^{-1} (x_2 - \overline{x}_2). \tag{141}$$

and

$$\mathcal{S}_B = \frac{1}{2}\left[\sum_{\alpha=1}^{N-2} \Xi_{\alpha N}^T [\Sigma_{NN}]^{-1} \Xi_{\alpha N} + \begin{pmatrix} \Xi_{N-1N} \\ \Xi_{NN} + p(p-1)\epsilon \end{pmatrix}^T [\Sigma_{\{1,0\}}]^{-1} \begin{pmatrix} \Xi_{N-1N} \\ \Xi_{NN} + p(p-1)\epsilon \end{pmatrix}\right],$$
(149)

we see that the quantity $\mathcal{V}$ in (126) equals to

$$\mathcal{V} = \mathcal{S}_0 - \mathcal{S}_B - \frac{1}{2}\sum_{\alpha\leq\beta=1}^{N-2}\sum_{\gamma\leq\delta=1}^{N-2}\sum_{a,b=1}^{2}\left(\Xi_{\alpha\beta}^a + m_{\alpha\beta}^a\right)[\Omega_0]_{\alpha\beta,\gamma\delta}^{ab}(\Xi_{\gamma\delta}^b + m_{\gamma\delta}^b) - \frac{1}{2}\times$$
$$\left[\sum_{\alpha=1}^{N-2}(m_{\alpha N-1} + \Xi_{\alpha 1}^*)^T[\Sigma_{\frac{1}{2}}^*]^{-1}(m_{\alpha N-1} + \Xi_{\alpha 1}^*) + (m_{N-1N-1} + \Xi_{11}^{**})^T[\Sigma_1^*]^{-1}(m_{N-1N-1} + \Xi_{11}^{**})\right].$$
(150)

Substituting this expression into (125) we obtain

$$\mathcal{I}(\epsilon_2,q|\epsilon_1) \propto \int_{s_1\cdot s_2=Nq} ds_1 ds_2 \int_{s_2(0)=s_2} \mathcal{D}s^t \mathcal{D}\hat{s}^t \, e^{\mathcal{V}_0 + \mathcal{S}_0 - \mathcal{S}_B}\mathcal{K}[s^t,\hat{s}^t],$$
(151)

which coincides with Eq. (23) with the identification (24). The term $\mathcal{K}[s^t,\hat{s}^t]$ contains all the terms depending on the components $m_{\alpha\beta}^a$: its structure is described in detail in the following Appendix.

## C  The integral over the Hessian matrices

After shifting the integration variables $m_{\alpha\beta}^a$ and implementing the constraints, we see that the term $\mathcal{K}[s^t,\hat{s}^t]$ in (23) can be compactly written as

$$\mathcal{K}[s^t,\hat{s}^t] = \int\prod_{a=1}^{2} d\overline{m}^a \, e^{-\frac{1}{2}\sum_{\alpha\leq\beta=1}^{N-1}\sum_{\gamma\leq\delta=1}^{N-1}\sum_{a,b=1}^{2}\overline{m}_{\alpha\beta}^a[\Omega^*]_{\alpha\beta,\gamma\delta}^{ab}\overline{m}_{\gamma\delta}^b}\prod_{a=1}^{2}|\det(\overline{m}^a - \Phi^a[s^t,\hat{s}^t] - p\epsilon_a\mathbb{1})|,$$
(152)

where the components of the $(N-1)\times(N-1)$ matrices $\overline{m}^a$ are given in the particular bases $\mathcal{B}_a$ introduced in Appendix B. It follows from (152) that the entries of $\overline{m}^a$ are Gaussian variables, with covariance matrix having the following structure:

$$[\Sigma^*]_{\alpha\beta,\gamma\delta}^{ab} = \delta_{\alpha\gamma}\delta_{\beta\delta}\left(\chi_{\alpha,\beta\leq N-2}(1+\delta_{\alpha\beta})[\Sigma_0^*]^{ab} + \chi_{\alpha\leq N-2}\delta_{\beta N-1}[\Sigma_{1/2}^*]^{ab} + \delta_{\alpha N-1}\delta_{\alpha\beta}[\Sigma_1^*]^{ab}\right),$$
(153)

where the $\Sigma_i^*$ are $2\times 2$ matrices computed explicitly in Appendix B. In particular,

$$\Sigma_0^* = \frac{p(p-1)}{2N}\begin{pmatrix} 1 & q^{p-2} \\ q^{p-2} & 1 \end{pmatrix}, \quad \Sigma_{1/2}^* = \frac{p(p-1)}{2N}\begin{pmatrix} 1 - \frac{(p-1)(1-q^2)q^{2p-4}}{1-q^{2p-2}} & -\frac{q^{2p}-(p-1)q^4+(p-2)q^2}{q^{p+3}-q^{5-p}} \\ -\frac{q^{2p}-(p-1)q^4+(p-2)q^2}{q^{p+3}-q^{5-p}} & 1 - \frac{(p-1)(1-q^2)q^{2p-4}}{1-q^{2p-2}} \end{pmatrix}.$$
(154)

Each of two matrices $\overline{m}^a$ is therefore made of an $(N-2)\times(N-2)$ block of entries having a GOE statistics that is basis invariant; every entry $\overline{m}_{\alpha\beta}^a$ in this block is correlated only with itself, and with the analogous entry $\overline{m}_{\alpha\beta}^b$ of the other matrix. This remains true also for the entries belonging to the last line and column of the matrices $\overline{m}^a$: their correlations, however, are different; moreover, even their variance depends explicitly on the overlap $q$ [10].

---

[10]This is due to the fact that we are expressing the components of each matrix in a basis in which only the $(N-1)$-th vector has an overlap with the vectors $s_1$ and $s_2$, see Appendix B.

We now come to the shifts $\Phi^a[s^t, \hat{s}^t]$ in (152). It follows from the derivation in Appendix B that these are $(N-1) \times (N-1)$ symmetric matrices with components:

$$\Phi^a_{\alpha\beta}[s^t, \hat{s}^t] = \chi_{\alpha,\beta \leq N-2}\, \Xi^a_{\alpha\beta} + \chi_{\alpha \leq N-2}\delta_{\beta,N-1}[\Xi^*_{\alpha 1}]^a + \delta_{\alpha,N-1}\delta_{\beta,N-1}[\Xi^{**}_{11}]^a. \tag{155}$$

A simple calculation gives:

$$\begin{aligned}
\Xi^a_{\alpha\beta} = {} & \frac{p(p-1)}{2N} \int_0^\infty dt [c_{a2}(0,t)]^{p-2} \left( [\hat{s}_2(t)]_\alpha [s_2(t)]_\beta + [s_2(t)]_\alpha [\hat{s}_2(t)]_\beta \right) \\
& + \frac{p(p-1)(p-2)}{2N} \int_0^\infty dt [c_{a2}(0,t)]^{p-3} r_{a2}(0,t)[s_2(t)]_\alpha [s_2(t)]_\beta,
\end{aligned} \tag{156}$$

where we used the notation $c_{ab}(t',t) = s_a(t') \cdot s_b(t)/N$ and $r_{ab}(t',t) = s_a(t') \cdot \hat{s}_b(t)/N$. We recall that the components of $\Xi^1_{\alpha\beta}$ are given in the basis $\mathcal{B}_1$, and those of $\Xi^2_{\alpha\beta}$ in the basis $\mathcal{B}_2$. Performing the necessary algebra we find:

$$\begin{aligned}
[\Xi^*_{\alpha 1}]^a &= c^a_1 \Xi^a_{\alpha N-1} + c^a_2 \Xi^a_{\alpha N}, \\
[\Xi^{**}_{11}]^a &= d^a_1 \Xi^a_{N-1 N-1} + d^a_2 \Xi^a_{N-1 N} + d^a_3 \Xi^a_{NN} - \begin{pmatrix} \Sigma_{N-1 N-1, N-1 N} \\ \Sigma_{N-1 N-1, NN} \end{pmatrix}^T [\Sigma_{\{1,0\}}]^{-1} \begin{pmatrix} 0 \\ p(p-1)\epsilon \end{pmatrix},
\end{aligned} \tag{157}$$

where and $c^a_x, d^a_{xy}$ are constants (depending on the overlap parameter $q$). Therefore the matrices $\Phi^a$ for $a = 1, 2$ have components given by:

$$\begin{pmatrix} \Phi^1_{\alpha\beta} \\ \Phi^2_{\alpha\beta} \end{pmatrix} = \begin{pmatrix} \mathcal{L}_1\left(\{\Xi^1_{\alpha'\beta'}\}\right) \\ \mathcal{L}_2\left(\{\Xi^2_{\alpha'\beta'}\}\right) \end{pmatrix} - \delta_{\alpha,N-1}\delta_{\beta,N-1} \begin{pmatrix} \Sigma_{N-1 N-1, N-1 N} \\ \Sigma_{N-1 N-1, NN} \end{pmatrix}^T [\Sigma_{\{1,0\}}]^{-1} \begin{pmatrix} 0 \\ p(p-1)\epsilon \end{pmatrix}, \tag{158}$$

with $\mathcal{L}_a$ linear functions of their arguments. The second term takes the explicit form:

$$\begin{pmatrix} \Sigma_{N-1 N-1, N-1 N} \\ \Sigma_{N-1 N-1, NN} \end{pmatrix}^T [\Sigma_{\{1,0\}}]^{-1} \begin{pmatrix} 0 \\ p(p-1)\epsilon \end{pmatrix} = \begin{pmatrix} \mu_1(q,\epsilon_1,\epsilon_2) \\ \mu_2(q,\epsilon_1,\epsilon_2) \end{pmatrix} = \frac{1}{a_2(q)} \begin{pmatrix} \epsilon_2 a_0(q) - \epsilon_1 a_1(q) \\ \epsilon_1 a_0(q) - \epsilon_2 a_1(q) \end{pmatrix}, \tag{159}$$

with the same functions as in (38). This implies that

$$\Phi^a_{\alpha\beta} = \phi^a_{\alpha\beta}[s^t, \hat{s}^t] - \delta_{\alpha,N-1}\delta_{\beta,N-1}\, \mu_a(q,\epsilon_1,\epsilon_2), \tag{160}$$

as stated in Eq. 36, where $\phi^a_{\alpha\beta}[s^t, \hat{s}^t] = \mathcal{L}_a\left(\{\Xi^a_{\alpha'\beta'}\}\right)$ is a linear combination of the integrals (156).

Equipped with these explicit expression, we can discuss the role of causality in the simplification of this term. The integrals (156) involve either products of the spin variable $s_2(t)$ and of the response field $\hat{s}_2(t)$ evaluated exactly at the *same* time, or terms proportional to the response function $r_{a2}(0,t)$. When the dynamical evolution is causal, these terms will typically be equal to zero: therefore, when the average over dynamical trajectories is restricted to causal ones, we can set $\phi = 0$. This simplifies considerably the shifts $\Phi^a$, that reduce to simple rank-1 projectors. Exploiting this crucial observation, we finally obtain:

$$\begin{aligned}
\mathcal{K} \overset{\text{causality}}{\longrightarrow} & \int \prod_{a=1}^2 d\overline{m}^a\, e^{-\frac{1}{2}\sum_{\alpha \leq \beta=1}^{N-1}\sum_{\gamma \leq \delta=1}^{N-1}\sum_{a,b=1}^2 \left(\overline{m}^a_{\alpha\beta} - \delta_{\alpha N-1}\delta_{\alpha\beta}\mu_a\right)[\Omega^*]^{ab}_{\alpha\beta,\gamma\delta}\left(\overline{m}^b_{\gamma\delta} - \delta_{\gamma N-1}\delta_{\gamma\delta}\mu_b\right)} \times \\
& \times \prod_{a=1}^2 |\det(\overline{m}^a - p\epsilon_a \mathbb{1})|.
\end{aligned} \tag{161}$$

By direct comparison with the the results recalled in Appendix A, we see that the matrix $\overline{m}^2$ in (161) reproduces exactly the statistics as the conditional Hessian matrices at a stationary point $s_2$ at fixed overlap $q$ from a reference minimum $s_1$, as expected. More precisely, $\sqrt{2N}\overline{m}^2 = \mathcal{M}$ with $\mathcal{M}$ defined in (110). The symmetric statement holds for $\overline{m}^1$. This allows us to conclude that (39) holds true.

# D Derivation of the boundary terms in the action

In this Appendix we derive the boundary terms in (29). The first term is given by

$$S_B^{(1)} = \frac{1}{2} \sum_{\alpha=1}^{N-2} \Xi_{\alpha N}^T [\Sigma_{NN}]^{-1} \Xi_{\alpha N}. \tag{162}$$

This term arises from conditioning the points $s_1$ and $s_2$ to be stationary points: in fact, it emerges from the constraint $m_{\alpha N}^a = 0$, which corresponds to setting the gradients to zero. From (136) we find that:

$$[\Sigma_{NN}]^{-1} = \frac{2N}{p(p-1)^2} \frac{q^2}{(q^2 - q^{2p})} \begin{pmatrix} 1 & -q^{p-1} \\ -q^{p-1} & 1 \end{pmatrix}. \tag{163}$$

Moreover, with the notation introduced in (26) we find:

$$\sum_{\alpha=1}^{N-2} \Xi_{\alpha N}^a \Xi_{\alpha N}^b =$$
$$\frac{p^2(p-1)^2}{4} \int_0^\infty dt \int_0^\infty dt' \left\{ \left[ c_{a2}(t) c_{b2}(t') \right]^{p-1} \overline{d}_{22}(t,t') + \left[ c_{a2}(t) c_{b2}(t') \right]^{p-2} r_{a2}(t) r_{b2}(t') \overline{c}_{22}(t,t') \right\}$$
$$+ \frac{p^2(p-1)^2}{4} \int_0^\infty dt \int_0^\infty dt' (p-1) \left[ c_{a2}(t) c_{b2}(t') \right]^{p-2} \left\{ r_{a2}(t) c_{b2}(t') \overline{r}_{22}(t,t') + c_{a2}(t) r_{b2}(t') \overline{r}_{22}(t',t) \right\}, \tag{164}$$

where

$$\overline{c}_{22}(t,t') = c_{22}(t,t') - \frac{c_{12}(t) c_{12}(t') - q c_{22}(t) c_{12}(t') - q c_{12}(t) c_{22}(t') + c_{22}(t) c_{22}(t')}{1-q^2}$$
$$\overline{d}_{22}(t,t') = d_{22}(t,t') - \frac{r_{12}(s) r_{12}(t') - q r_{22}(t) r_{12}(t') - q r_{12}(t) r_{22}(t') + r_{22}(t) r_{22}(t')}{1-q^2}$$
$$\overline{r}_{22}(t,t') = r_{22}(t,t') - \frac{c_{22}(t) r_{22}(0,t') - q c_{12}(0,t) r_{22}(0,t') - q c_{22}(t) r_{12}(0,t') + c_{12}(0,t) r_{12}(0,t')}{1-q^2}. \tag{165}$$

Combining everything we get the expression (30). The second contribution to the boundary terms is given by:

$$S_B^{(2)} = \frac{1}{2} \begin{pmatrix} \Xi_{N-1N} \\ \Xi_{NN} + p(p-1)\epsilon \end{pmatrix}^T [\Sigma_{\{1,0\}}]^{-1} \begin{pmatrix} \Xi_{N-1N} \\ \Xi_{NN} + p(p-1)\epsilon \end{pmatrix}. \tag{166}$$

This arises from conditioning on both the gradient and the energy density of the points $s_a$. In this case no summation over the indices has to be performed, and the expression (33) is obtained setting $A = [\Sigma_{\{1,0\}}]^{-1}$. Explicitly we find:

$$A = \mathcal{A}(q) \begin{pmatrix} A^{(1)} & A^{(2)} \\ A^{(2)} & A^{(3)} \end{pmatrix}, \tag{167}$$

with

$$\mathcal{A}(q) = \frac{1}{p(p-1)[q^{4p} - ((p-1)^2 q^4 - 2(p-2)pq^2 + (p-1)^2) q^{2p} + q^4]}, \tag{168}$$

and

$$A^{(1)} = \begin{pmatrix} q^4 - q^{2p}\left(p\left((p-1)q^4 + (3-2p)q^2 + p - 2\right) + 1\right) & q^{3p}\left(p\left(q^2-1\right)+1\right) - q^{p+4} \\ q^{3p}\left(p\left(q^2-1\right)+1\right) - q^{p+4} & q^4 - q^{2p}\left(p\left(q^2-1\right)\left((p-1)q^2 - p + 2\right) + 1\right) \end{pmatrix}$$

$$A^{(2)} = \begin{pmatrix} (p-1)pq^{2p+1}\left(1-q^2\right)^{3/2} & -pq^{p+1}\sqrt{1-q^2}\left(q^2-q^{2p}\right) \\ -pq^{p+1}\sqrt{1-q^2}\left(q^2-q^{2p}\right) & (p-1)pq^{2p+1}\left(1-q^2\right)^{3/2} \end{pmatrix}$$

$$A^{(3)} = \begin{pmatrix} pq^4 - pq^{2p+2}\left(-pq^2+q^2+p\right) & -pq^{p+2}\left(q^{2p}-pq^2+p-1\right) \\ -pq^{p+2}\left(q^{2p}-pq^2+p-1\right) & pq^4 - pq^{2p+2}\left(-pq^2+q^2+p\right) \end{pmatrix}.$$

(169)

# E  Constants appearing in dynamical equations

Let us introduce:

$$D(q) = q^{4p} - \left((p-1)^2 q^4 - 2(p-2)pq^2 + (p-1)^2\right)q^{2p} + q^4. \tag{170}$$

The constants appearing in the equation for the overlap $x(t)$ read:

$$
\begin{aligned}
D(q)G_1^1(q) &= pq^{p+1}\left((p-2)q^{2p} - (p-1)q^4 + q^2\right) \\
D(q)G_1^2(q) &= -pq\left[\left((p-3)p\left(q^2-1\right)+q^2-2\right)q^{2p} + q^4\right] \\
D(q)G_2^1(q) &= -p\left(q^4 - q^{2p}\left(\left(-p^2+p+1\right)q^2+(p-1)q^4+(p-1)^2\right)\right) \\
D(q)G_2^2(q) &= (2-p)pq^{p+4} + p\left(p-q^2-1\right)q^{3p} \\
D(q)G_3^1(q) &= (p-1)pq^{p+2}\left(q^2-q^{2p}\right) \\
D(q)G_3^2(q) &= (p-1)^2 p\left(q^2-1\right)q^{2p+2} \\
D(q)G_4^1(q) &= (p-1)^2 p\left(q^2-1\right)q^{2p+1} \\
D(q)G_4^2(q) &= (p-1)pq^{p+1}\left(q^2-q^{2p}\right),
\end{aligned}
$$

(171)

while those appearing in the equation for the correlation are given by:

$$
\begin{aligned}
D(q)F_1^1(q) &= (p-1)pq^p\left(q^{2p}-q^4\right) \\
D(q)F_1^2(q) &= -p\left(\left((p-2)p\left(q^2-1\right)-1\right)q^{2p}+q^4\right) \\
D(q)F_2^1(q) &= (p-1)p\left(q^2-1\right)q^{2p+1}(p-1)p\left(q^2-1\right)q^{2p+1} \\
D(q)F_2^2(q) &= pq^{p+1}\left(q^2-q^{2p}\right) \\
D(q)F_3^1(q) &= (p-1)^2 p\left(q^2-1\right)q^{2p+1} \\
D(q)F_3^2(q) &= (p-1)^2 p\left(q^2-1\right)q^{2p+1} \\
D(q)F_4^1(q) &= -p\left(\left((p-2)p\left(q^2-1\right)-1\right)q^{2p}+q^4\right) \\
D(q)F_4^2(q) &= (p-1)pq^p\left(q^{2p}-q^4\right) \\
D(q)F_5^1(q) &= (p-1)^2 p\left(q^2-1\right)q^{2p+1} \\
D(q)F_5^2(q) &= (p-1)pq^{p+1}\left(q^2-q^{2p}\right) \\
D(q)F_6^1(q) &= pq^{p+1}\left(q^2-q^{2p}\right) \\
D(q)F_6^2(q) &= (p-1)p\left(q^2-1\right)q^{2p+1}.
\end{aligned}
$$

(172)

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
