# Peer review of "Dynamical Instantons and Activated Processes in Mean-Field Glass Models"

_SciPost Physics, doi:SciPost Phys. 10, 002 (2021)_

## Round 2 · Referee Report · Giampaolo Folena (Referee 1) · 2020-8-31

Report

This article further deepens the knowledge on activated events in complex mean-field energy landscapes, giving new mathematical tools that may open the path to new discoveries in the field. The focus is on the pure (homogeneous) p-spin spherical, which in the last years has received an increasing interest within both the mathematical and physical community. The main result is the possibility of geometrically connecting two minima of the energy landscape through a saddle and deriving the mean-field “dynamical instantons” equations that control this barrier jumping. In my opinion, the greatest achievement consists in evaluating the dynamical mean-field equations for correlation and response, given some highly-non-trivial conditioning, i.e. initiating the system in a given saddle at a given distance from a reference minimum.

The Abstract is clear and exhaustive, however, I would mention that the analyzed model is the pure (homogeneous) p-spin. The fact that the pure model is considered, allows to rewrite the mean-field action in terms of matrix elements and therefore allows the simplification used when conditioning (Eq. 18 and 19). Moreover, all the general comments about the energy landscape and the presence of an energy threshold are valid concepts only in pure models.

  1. The introduction is very well written. Nevertheless, some small comments are needed. The phrase “mean-field models displays two dynamical regimes: [...]approaches ( or more precisely ages toward) the threshold states. [...]” is misleading since in general there is no unique threshold. Again the homogeneous p-spin models are not generic mean-field models. Moreover, when introducing the dynamical instanton, I would be more careful to precisely introduce the idea, since, for what I have grasped, it is a new concept; and in the phrase “This is different with respect to standards phase transition […]”, maybe instead of ‘standard’ the word ‘equilibrium’ can be used? It is the concept of equilibrium and out-of-equilibrium connected to the definition of dynamical instanton?

  2. The Summary of results here all the main results are presented in a very simple form, without technicalities. In fig.3 I would suggest some isoclines to better see the variations. Still, I am a bit confused by the dynamical instanton definition. Shouldn’t the correlation c(t,t’) plotted in fig.4 depend on the dimension N of the system? Does the temperature enter in the definition of dynamical instanton?

  3. Self-consistent … this is the core of the article where the main analytical results are derived. Here the mean-field dynamical equations for correlation and response of the dynamics conditioned to start from a saddle at a given distance from a reference minimum are derived. The calculations presented are not self-contained and some results were derived by the same authors in previous articles. The choice on how to split the calculations between section 3 and the appendices is a bit arbitrary, and in my opinion, the reader cannot easily follow the steps going back and forth. I would suggest to directly move the formulas presented in subsection 3.1.3 in the appendices and just give a comment on the term K(s,s^) in the main text. In page 14 “A standard calculation gives:” would be nice to have a reference. In equation 27 an integral is missing. In equation 29, S(2) → S(1). And maybe would be better to use t,t’ for times variables Section 3.3 is clear. I think that this is the most important part of this work and I would have taken more care in the explanation of the calculations in the hope that this could be used as a starting point for many other similar calculations by other researchers in the field. On the other hand, I understand the technical difficulties to explain in details such a calculation. Therefore, this is only a suggestion.

  4. Where does the system fall… this section presents the asymptotic analysis of the conditioned mean-field equations making use of the TTI and FDT assumptions (stationary limit), allowing to connect the reference saddle with the two minima on the two side of the barriers. This is a standard calculation and the results are very well exposed. I suggest to write down the energy of the first minimum in all the figures. I appreciated fig. 7 (right). Perfect.

  5. Numerical solution … this section confirms the asymptotic results obtained in section 4. The kick procedure is clear at the analytical level, however, I had some difficulty following the numerical implementation of the kick. In subsection 5.2 it is not true that to evaluate c(t,t’) one needs only c(t-dt,s) for all s<t-dt, but rather every point c(s,s’) with s<t’ and s<t. The subsequent discussion on the kick implementation is not quite clear. Also, I would cite some previous work on the numerical implementation of the integration scheme. The last phrase in parenthesis “(only the latter has a non-trivial TTI dynamics since a=0)”, what does it mean?

  6. The shape of the dynamical instanton ... despite the fact that this section defines the title of the article, to me it seems less important than the previous ones. The dynamical instanton is evaluated by reversing the downhill dynamics from the saddle and joining it with the dynamics from the saddle to the second minimum. The final result is not the exact instanton but rather an approximation, which merges the two-time sectors. In Eq. 102 and 103 \tau_S → \tau_s One point that I think is not clear is how in Fig. 4 the \tau_s, that defines the time of saddle crossing, is chosen. Is \tau_s chosen arbitrarily? and if that is the case, what is the relative dimension N and temperature at which this instantonic solution would be expected to give a contribution to the equilibrium dynamics in a finite-size system?

  7. Conclusion It is maybe the less impressive part of this work. It gives a good resumption of the article but does not contribute to a concrete perspective that these calculations will permit. In particular, it seems that all the efforts in this work bring only a qualitatively understanding: “whereas the ones analysed in this work are more likely to give rise to back and forth motions with frequent returns to the original minimum.” How can the dynamical instanton give some insights into the equilibrium dynamics of a finite size model?

My general opinion is that this article makes fundamental steps forward in the field of mean-field dynamical equations, and activated events in complex landscapes. The analysis is mainly devoted to mathematical aspects rather than the physical ones. The results are very powerful methods to deal with conditioned dynamics; in my opinion, it only lacks the physical perspective a bit. I have appreciated the effort on introducing the subject in the first two sections with very clear and technicality-free literature. I would suggest some more care in the introduction of dynamical instanton and the physical motivations behind its calculation.

---

## Round 2 · Referee Report · Anonymous (Referee 2) · 2020-9-22

Referee report on the paper "Dynamical Instantons and Activated Processes in Mean-Field Glass Models"

by V. Ros, G. Biroli and C. Cammarota

This is an excellent and deep paper which sheds the new and constructive light on a long-standing problem in theory of disordered systems: description of an activated escape from local minimum via barrier-crossing in mean-field models with one-step replica symmetry breaking mechanism. The corresponding dynamics probes time-scales exponentially large in the size $N$ of the system. The correct order parameter allowing to describe glassy dynamics on such scales is a two-time correlation function, which is a quite difficult and less explored object to study. Important role in building the theory is played by a clever combination of Kac-Rice framework of counting stationary points with a fixed overlap with a given minimum with dynamical equations describing the evolution of the system. The authors derive the dynamical mean-field equations corresponding to Langevin dynamics, with initial condition chosen to be precisely at one (out of many) index-one saddles in the vicinity of local minimum. They then find a convincing way of analysing them both numerically and (to an extent) analytically. As a result, they provide quite a detailed picture explaining how activated crossing happens through saddles at different overlap and energy density. Along the way the authors show how to obtain the dynamical instanton corresponding to the involved activated process, and the associated shape of the two-point correlation function (which turns out to be quite involved). Technically the analysis is quite involved, and performing it is a considerable achievement. One of conclusions of the paper is that escape through most numerous saddles with big overlap leads typically to minima at higher energy. this indirectly implies that descrease in energy occurs through less numerous sadles which have smaller overlap, and their studies is left for future work.

In summary, this is an important work essentially contributing to our understanding of thermal relaxation in mean-field glassy models, and I strongly recommend its publication.

---

## Round 2 · Referee Report · Anonymous (Referee 3) · 2020-10-13

Strengths

Technical paper of very high level. This is a (very valuable) step towards a very difficult task.

Weaknesses

That it does not solve the full problem, but that is very very hard!

Report

The authors build upon results on the free-energy landscape of disordered p-spin models that they derived in recent years, to make a step towards the (very difficult) analytical study of thermally activated processes in models with rugged free-energy landscapes. More precisely, they obtain the dynamical instanton corresponding to a particular process, the passage from a local minimum to another one via a saddle of rank-1 in the spherical p-spin disordered model. This is a very interesting and useful contribution to the theory of the dynamics of complex systems and it merits publication in SciPost. Below I list some changes that I think the authors should implement in the preprint before its publication.

Having chosen a minimum, the saddle and the other minimum, where is the difference with a “normal” instanton calculation? I think the authors should stress in which sense this calculation is different and what makes it special.

The dynamics one is trying to characterise are the ones in a rugged free-energy landscape. In the studies in which the initial states were drawn from the Boltzmann equilibrium pdf, this was ensured. Here, the initial states are chosen with respect to the potential energy landscape. This simplification, that one can accept in the pure p-spin model due to the simplicity of the temperature effects on the free-energy landscape, can diminish the relevance of this calculation for other disordered models. The authors should discuss this important point somewhere in the text. (If they have already done it, I missed it.)

The authors might find useful to compare their results to the simulations in D. A. Stariolo and L. F. Cugliandolo, Phys. Rev. E 102, 022126 (2020) where the MC dynamics of the Ising (instead of spherical) p-spin model was studied and properties of subsequent minima, their overlap, and the barriers crossed, were studied. In particular, the back and forth motion between nearby minima was observed numerically in this paper.

Notation requirements: It would be convenient to call m the minima and _s the saddles. Following the subindices is hard, very quickly one gets to the point of asking one-self who was _1, who was _2? Moreover, the choice of \infty is not happy either, I would say. Instead, Fig. 1 is very useful indeed. In eq. (29) the integrals over s and s’ should be integrals over times, right? So please use time notation, s was used for something else beforehand. The same in App. D and the following equations. It’s a pity the overflows in eq. (49) and maybe also in other equations, please check and arrange.

Sec. 2.2 Through the saddles -> through the saddle, right? Because the saddle was also chosen. A clarification of this point is needed.

The fact that the response equation [eq. (55) in the preprint] is ``resilient’’ to changes in the initial conditions, etc. is a quite common fact. The authors could comment on this.

A grammar check would be welcome. I spotted a few English mistakes such as: - Systems jumps -> systems jump in page 2 one of the main aim -> one of the main aims in page 6 not clear what is the need for the phrase “- an information that is missing so far” also in page 6. Is ``to transition’’ a verb? Page 19. Surely there are others.

Requested changes

See the report.

---

## Round 3 · Author Response

Dear Editor,

We would like to thank you for handling our manuscript, and to thank the Referees
for their careful reading of the manuscript and for their constructive comments to improve its quality. We have revised the manuscript incorporating the changes suggested in the reports. We submit here the updated version, and provide below the answers to the Referee’s comments and questions. The changes performed on the manuscript are listed within the replies to the Referees.

Yours sincerely,
The authors

 <b>Answers to Referee 1 </b>

 We are grateful to the Referee for his very careful reading of our manuscript, and for the very thorough comments on the work and on the results. Below, we reply to the Referee's comments and observations in the order in which they are given in the Report. 

<ul>

<li><b> <em> The Abstract is clear and exhaustive, however, I would mention that the analyzed model is the pure (homogeneous) p-spin.</em></b> 

We thank the Referee for this remark; in order to stress that the analysis in restricted to the pure case, we specified this in the abstract as suggested, as well as in proximity to the Equations 19 and 20.  </li>

<li><b> <em> The phrase “mean-field models displays two dynamical regimes: […] approaches (or more precisely ages toward) the threshold states. [...]” is misleading since in general there is no unique threshold. Again the homogeneous p-spin models are not generic mean-field models.</em></b> 

We thank the Referee for this comment. In the original version of the manuscript, we referred to the threshold states as those marginally stable states that are reached asymptotically by the dynamics starting from initial conditions of high enough energy. In the revised version, we clarified this point on page 2 by stressing that the initial condition corresponds to high enough temperature, and added a footnote to recall the more general setup described, for instance, in what is now cited as Ref. [9]. In addition, we modified a sentence on page 2 into <em>"the system approaches (or more precisely ages toward) the threshold (or more generally, the marginally stable) states"</em>. </li>

<li><b> <em> Moreover, when introducing the dynamical instanton, I would be more careful to precisely introduce the idea, since, for what I have grasped, it is a new concept; and in the phrase “This is different with respect to standards phase transition […]”, maybe instead of ‘standard’ the word ‘equilibrium’ can be used? It is the concept of equilibrium and out-of-equilibrium connected to the definition of dynamical instanton?</em></b> 

In order to clarify how instantons are obtained in the case of simpler stochastic processes and in which way this approach differs from our calculation, we have added a few comments in Sec. 2.3. Regarding the second part of the Referee's comment, we agree that the mention "standard phase transitions" could be misleading (in particular, it did not refer to equilibrium vs non equilibrium); the purpose of that sentence was to stress the difference between the case of glasses (where the order parameter is a two-point function) and the general scenario of phase transitions, where the order parameter usually depends on one time only. We rephrased that sentenced as follows: <em>"[…] the correct order parameter that describes glassy dynamics is the correlation function between two different times. This introduces an additional degree of difficulty with respect to the standard setting in phase transitions, where the order parameters are typically one-time (or point) functions."</em> </li>

<li><b> <em> In fig.3 I would suggest some isoclines to better see the variations. </em></b> 

We considered this nice suggestion: instead of the isoclines, we actually found more insightful to plot the level curves in this Figure. </li>

<li><b> <em> Shouldn’t the correlation c(t,t’) plotted in fig.4 depend on the dimension N of the system? Does the temperature enter in the definition of dynamical instanton? </em></b> 

We agree that it is very important to clarify how <em>N</em> and <em>T</em> enter in our definition. Throughout our treatment <em>N</em> is assumed to be very large: the correlation <em>c(t,t')</em> does not depend explicitly on the system dimension <em>N</em>, as it is obtained minimizing the term in the dynamical action that depends extensively on <em>N</em>. The dependence on temperature is encoded in the parameter α in the dynamical equations, which is set to zero for the numerical resolution of the equations. In order to clarify further these points, we added the following comments in the conclusion in the revised version of the manuscript: <em>"Let us comment on the role of the dimensionality N and of the temperature T in our analysis. As we have stressed in Sec. 2.3, our calculation differs with respect to a standard instanton calculation performed minimizing a suitably-defined large deviation (dynamical) functional.
We use the knowledge gained from the Kac-Rice analysis about the index-1 saddles around one given minimum and study the relaxation dynamics starting from a given saddle.The relaxation from the saddle to the minima is a typical dynamical process, and thus it does not require to compute large deviations of the dynamical functionals.
The instanton is then obtained by time-reversing the solution
which goes back to the original minimum.
This approach allows us to obtain insights on the dynamics at times scaling exponentially with the system size <em>N</em> and does not require to solve the very challenging problem of finding non-casual solutions associated to the large deviation (dynamical) functional [36].
Note that in our analysis we do not take into account the finite <em>N</em> corrections to the landscape statistics; analyzing how those affect dynamics at finite times remains a challenging open question.
For what concerns the role of temperature, the initial conditions of our dynamical equations are unstable stationary points of the energy landscape; moreover, we solved the dynamical equations setting α=0, thus describing gradient descent from the saddle to a nearby minimum of the energy landscape. The instantons we obtain are therefore expected to capture the dynamics at very small values of temperature: small enough so that the energy landscape is a good approximation of the free-energy one, but non-zero, so that barrier-crossing is a possible even though extremely rare process. This is particularly meaningful for the spherical p-spin model, given that the free-energy landscape has a continuous dependence on temperature in that model.
For more generic models, the relevant landscape at finite temperatures is the free-energy one. In the fully-connected limit, the free-energy landscape of the system can be characterized in terms of the so called TAP functional f<sub>TAP</sub>(<b>m</b>) [48], depending on the local magnetizations <b>m</b>=(m<sub>1</sub>, ..., m<sub>N</sub>) rather than on the spin configurations <b>s</b>.
The stationary points of this functional have a physical meaning: stable local minima can be identified with the system’s metastable states; unstable saddles are also attractors of the dynamics, when thought of as an evolution on the free energy surface [49]. Therefore, one can envisage a dynamical calculation similar to the one presented in this work, with the TAP free energy replacing the energy landscape. The special instantonic solution found in [19] shows in a concrete example that using the TAP landscape to analyze thermal activation is justified.
Note that this finite temperature treatment could be particularly interesting to pursue for models in which thermal fluctuations modify substantially the landscape giving rise to a chaotic dependence on temperature [50,51,52] (see Ref. [53] for recent progress in the characterization of these landscapes)." </em> 

References [49-53] have been added.
</li>

<li><b> <em> The choice on how to split the calculations between section 3 and the appendices is a bit arbitrary, and in my opinion, the reader cannot easily follow the steps going back and forth. I would suggest to directly move the formulas presented in subsection 3.1.3 in the appendices and just give a comment on the term K in the main text. In page 14 “A standard calculation gives:” would be nice to have a reference.</em></b> 

We thank the Referee for this suggestion. We are aware that, as the Referee points out, Section 3.1.3 is particularly technical and not easy to follow.
After thinking about how to redistribute the content, we have decided not to reduce its size as suggested, but to limit its content to the explicit expression of the dynamical action, which is the starting point for the derivation of the dynamical equations through functional variation. In the updated version of the manuscript, we have added a few comments in section 3.2 which (we hope) can guide the reader willing to reproduce the calculations. Moreover, we have added a reference for the computation of the Gaussian integral over the dynamical variables s(t), ŝ(t) on pag.14. </li>

<li><b> <em> In equation 27 an integral is missing.
In equation 29, S(2) -> S(1). And maybe would be better to use t,t’ for times variables. I suggest to write down the energy of the first minimum in all the figures.</em></b> 

We have fixed the typos spotted by the Referee, and denoted all time quantities with <em> t, t'</em> or <em> t''</em>, reserving the notation <em>s</em> to the spin configurations only. We also thank the Referee for the suggestion about the figures, that we implemented. </li>

<li><b> <em> In subsection 5.2 it is not true that to evaluate c(t,t’) one needs only c(t-dt,s) for all s<t-dt, but rather every point c(s,s) with s<t’ and s<t. The subsequent discussion on the kick implementation is not quite clear. </em></b> 

We thank the Referee for allowing us to clarify this point. A simple numerical evaluation of <em>c(t,t’) </em> needs <em>c(t-dt,s) </em> and <em>c(t’,s)</em> for <em>s<t </em>, not all <em>c(s,s’) </em>, see for instance integral on <em>s</em> of <em>r(t,s) c(t,s)<sup>p-2</sup> c(t’,s)</em>. More refined algorithms might resort to iterative solutions of the equations which include all terms <em>c(s,s’)</em>, but we did not use them as it can be seen by looking at the reference cited for the algorithm. Also, to select properly the initial conditions, <em>c(t,0)</em> and <em>c(t’,0)</em> are needed but they are not specially affected by the kick, so we do not discuss them. In the revised version of the manuscript, we added <em>“among other terms”</em> to highlight that we specially discuss relevant terms only. We reworded the following explanation to improve clarity on the kick implementation. </li>

<li><b> <em> I would cite some previous work on the numerical implementation of the integration scheme.</em></b> 

We have added Refs. [44,45], cited as early works in which the numerical implementation of the dynamical equations is discussed. More recent works (e.g. Ref. [9]) are now cited in the introduction. </li>

<li><b> <em> The last phrase in parenthesis “only the latter has a non-trivial TTI dynamics since α=0”, what does it mean?</em></b> 

Since the temperature is zero, the TTI correlation function is simply equal to one at all times. Only the response function displays a non-trivial time-dependence. </li>

<li><b> <em> In Eq. 102 and 103 τ<sub>S</sub>-> τ<sub>s</sub>.
One point that I think is not clear is how in Fig. 4 the τ<sub>s</sub>, that defines the time of saddle crossing, is chosen. Is τ<sub>s</sub> chosen arbitrarily? and if that is the case, what is the relative dimension N and temperature at which this instantonic solution would be expected to give a contribution to the equilibrium dynamics in a finite-size system?</em></b> 

We thank the Referee for spotting the typo. Regarding the question on τ<sub>s</sub>, the latter is chosen arbitrarily with a dependence on the amplitude of the applied kick, as discussed at the beginning of section 5.1. All timescales related to barrier crossing diverge for <em>N</em> large.
For every finite <em>N</em> and <em>T</em> these instantonic solutions will allow ergodicity restoration at sufficiently large time as discussed in the introduction and at the beginning of section 6.2.
To make clearer the first point, at the beginning of section 6.2 we also added: <em>“In the large N limit, both these free-fall dynamics need infinite time, τ<sub>up</sub> and τ<sub>do</sub>, to take place, but thanks to the introduction of the kick they can be visualised in a finite time window.” </em> </li>

<li><b> <em> In particular, it seems that all the efforts in this work bring only a qualitatively understanding: “whereas the ones analysed in this work are more likely to give rise to back and forth motions with frequent returns to the original minimum.” How can the dynamical instanton give some insights into the equilibrium dynamics of a finite size model? I would suggest some more care in the introduction of dynamical instanton and the physical motivations behind its calculation.</em></b> 

As suggested by the Referee, in the revised version of the manuscript we have extended Sec. 2.3 and the Conclusions in order to introduce more extensively the notion of dynamical instanton, and to stress which type of hints on the activated dynamics follow from our analysis. In particular, what the analysis shows is that, given a reference minimum of the p-spin landscape, the saddles that are closer to it in configuration space connect the minimum to other minima at much higher energy. This suggests that if the system picks these saddles to escape from the minimum (which is likely to happen, at least in the shorter times of the dynamics), it will reach minima that are separated to the first one by an extensively smaller barrier than the one the system has just crossed. Therefore, it is likely that the system will return to the first minimum and attempt several escapes of this type, before managing to decorrelate completely from the reference minimum. In order to strengthen this expectation, we have added a reference to the numerical results in [47], where this type of motion has been observed in the direct simulation of the dynamics of an Ising p-spin model. </li>

</ul>

 <b>Answers to Referee 3 </b>

 We are very grateful to the Referee for deeming our work interesting and useful, and for recommending publication of the manuscript in Scipost. Moreover, we thank her/him for the relevant comments, inputs and observations. We reply to them in detail in the following. 

<ul>

<li><b> <em> Having chosen a minimum, the saddle and the other minimum, where is the difference with a “normal” instanton calculation? I think the authors should stress in which sense this calculation is different and what makes it special.</em></b> 

We are grateful to the Referee for stressing this point. In the revised version of the manuscript (in Sec. 2.3) we have added the following comment, that hopefully clarifies in which sense our approach differs from the standard instanton calculations: <em>"As sketched in Fig. 1, this solution is obtained conditioning the system to escape from s<sub>1</sub> through one particular, chosen index-1 saddle s<sub>2</sub>: it therefore does not represent the most general escape process, that should be obtained averaging over all possible dynamical trajectories connecting the two local minima, possibly through different saddles. We therefore expect that the c(t,t') in Fig. 4 represents a special solution of more general dynamical equations, obtained extremizing a large deviation functional as mentioned above."</em>  </li>

<li><b> <em> The dynamics one is trying to characterise are the ones in a rugged free-energy landscape. In the studies in which the initial states were drawn from the Boltzmann equilibrium pdf, this was ensured. Here, the initial states are chosen with respect to the potential energy landscape. This simplification, that one can accept in the pure p-spin model due to the simplicity of the temperature effects on the free-energy landscape, can diminish the relevance of this calculation for other disordered models. The authors should discuss this important point somewhere in the text.</em></b> 

We thank the Referee for raising this important point, and we agree with him/her on the fact that this should be discussed explicitly in the manuscript. In the revised version, we have added a discussion on the temperature dependence in the conclusion, in the following form: <em>"For what concerns the role of temperature, the initial conditions of our dynamical equations are unstable stationary points of the energy landscape; moreover, we solved the dynamical equations setting α=0, thus describing gradient descent from the saddle to a nearby minimum of the energy landscape. The instantons we obtain are therefore expected to capture the dynamics at very small values of temperature: small enough so that the energy landscape is a good approximation of the free-energy one, but non-zero, so that barrier-crossing is a possible even though extremely rare process. This is particularly meaningful for the spherical p-spin model, given that the free-energy landscape has a continuous dependence on temperature in that model.
For more generic models, the relevant landscape at finite temperatures is the free-energy one. In the fully-connected limit, the free-energy landscape of the system can be characterized in terms of the so called TAP functional f<sub>TAP</sub>(<b>m</b>) [48], depending on the local magnetizations <b>m</b>=(m<sub>1</sub>, ..., m<sub>N</sub>) rather than on the spin configurations <b>s</b>.
The stationary points of this functional have a physical meaning: stable local minima can be identified with the system’s metastable states; unstable saddles are also attractors of the dynamics, when thought of as an evolution on the free energy surface [49]. Therefore, one can envisage a dynamical calculation similar to the one presented in this work, with the TAP free energy replacing the energy landscape. The special instantonic solution found in [19] shows in a concrete example that using the TAP landscape to analyze thermal activation is justified.
Note that this finite temperature treatment could be particularly interesting to pursue for models in which thermal fluctuations modify substantially the landscape giving rise to a chaotic dependence on temperature [50,51,52] (see Ref. [53] for recent progress in the characterization of these landscapes)." </em> 

References [49-53] have been added.  </li>

<li><b> <em> The authors might find useful to compare their results to the simulations in D. A. Stariolo and L. F. Cugliandolo, Phys. Rev. E 102, 022126 (2020) where the MC dynamics of the Ising (instead of spherical) p-spin model was studied and properties of subsequent minima, their overlap, and the barriers crossed, were studied. In particular, the back and forth motion between nearby minima was observed numerically in this paper.</em></b> 

We thank the Referee for mentioning this work, which is referred to as Ref. [47] in the revised version of the manuscript. To compare with the numerical simulations, we have added the following comment in the conclusions:
<em>"We have found that the minima that are reached dynamically through these saddles are at higher energy than the reference one and are strongly correlated to it, being relatively close in configurations space. It is therefore natural to expect that escape processes through these saddles are likely to give rise to
back and forth motions of the system, with frequent returns to the original minimum.
Such frequent returns have been recently observed in numerical simulations of the low-temperature dynamics of the Ising p-spin model of finite-size [13, 47]. In [47] in particular it is shown that most of the stable configurations (the analogous of local minima in a discrete setting) that the system visits consecutively in its slow dynamics have a large overlap with each others; moreover, it appears that the system has to climb higher in the energy landscape in order to reach stable configurations that are less correlated with the previous one, consistently with our finding that the minima at smaller overlap with the reference one are connected to it by saddles at higher energy density (at least when focusing on the saddles at larger overlap and zero complexity, see Fig. 6)." </em>  </li>

<li><b> <em> The fact that the response equation [eq. (55) in the preprint] is
resilient to changes in the initial conditions, etc. is a quite common fact. The authors could comment on this.</em></b> 

We thank the Referee for this remark; in the revised version of the manuscript, we have added the following comment after the equation [now Eq. (57)] for the response function:
<em>"This equation is formally unaltered by the coupling to the initial conditions, the dependence on which is only implicit (through z(t) and c(t,t')). This is a generic feature, which occurs also whenever the initial conditions are extracted from a thermal measure [30,37]. It ultimately follows from the fact that the time evolution of the response function is governed by a memory kernel (the last term in Eq. 57) whose formal structure depends only on the gradient of the energy functional, and not on the configuration in which the system is initialized." </em>  </li>

</ul>

We also thank the Referee for the valuable suggestions on how to improve the readability of the manuscript. Below, we comment on those points and list the relative changes implemented in the manuscript.

<ul>
<li>  For what concerns the notation, the Referee suggests the relabeling s<sub>1</sub> -> s<sub>m</sub> and s<sub>2</sub> -> s<sub>s</sub>, in order to stress the fact that s<sub>1</sub> is a minimum of the energy landscape, while s<sub>2</sub> is typically chosen to be a saddle. Even though we understand the point of this observation, we decided to stick to the original notation; this is motivated by the fact that in our formalism the stationary point s<sub>2</sub> can be either a minimum (when the overlap parameter <em>q</em> is chosen small enough), or a saddle (for large enough <em>q</em>). Even though we are practically interested in the situation in which s<sub>2</sub> is a saddle, we prefer to keep the notation as general as possible, in particular not to generate confusion when discussing the stability of the "static" solution in Sec. 3.3.1.
We agree with the Referee on the fact that the notation s<sub>∞</sub> for the second minimum reached by the dynamics was not congruent: as a consequence, we made the replacement s<sub>∞</sub> -> s<sub>3</sub>, and relabeled the asymptotic values of the correlation and of the overlap with s<sub>1</sub> as indicated in the updated Fig. 1.  </li>

<li> In the updated version of the manuscript, all time quantities are denoted with <em>t, t'</em> and <em> t''</em>, and the notation <em>s</em> is reserved to the spin configurations only.  </li>

<li> In Sec. 2.2, we added the sentence <em>"we obtain the dynamical equations that allow to characterize the minimas s<sub>3</sub> that are connected to the reference one s<sub>1</sub> through one of the saddles s<sub>2</sub> lying in its vicinity"</em>, to stress the fact that both the reference minimum s<sub>1</sub> and the saddle s<sub>2</sub> are chosen.  </li>

<li> We made sure that no overflows is present in the revised version of the manuscript. We also thank the Referee for pointing out the several spelling and grammar mistakes, that we have corrected. </li>
</ul>

---

## Editorial Decision

published